# Deep Multi-Modal Structural Equations For Causal Effect Estimation With Unstructured Proxies

**Shachi Deshpande**[1,2], **Kaiwen Wang**[1,2], **Dhruv Sreenivas**[2], **Zheng Li**[1,2], **Volodymyr Kuleshov**[1,2]
Department of Computer Science, Cornell Tech[1] and Cornell University[2]
{ssd86, kw437, ds844, zl634, kuleshov}@cornell.edu

## Abstract

Estimating the effect of an intervention from observational data while accounting for confounding variables is a key task in causal inference. Oftentimes, the confounders are unobserved, but we have access to large amounts of additional unstructured data (images, text) that contain valuable proxy signal about the missing confounders. This paper argues that leveraging this unstructured data can greatly improve the accuracy of causal effect estimation. Specifically, we introduce deep multi-modal structural equations, a generative model for causal effect estimation in which confounders are latent variables and unstructured data are proxy variables. This model supports multiple multi-modal proxies (images, text) as well as missing data. We empirically demonstrate that our approach outperforms existing methods based on propensity scores and corrects for confounding using unstructured inputs on tasks in genomics and healthcare. Our methods can potentially support the use of large amounts of data that were previously not used in causal inference.

## 1 Introduction

An important goal of causal inference is to understand from observational data the causal effect of performing an intervention—e.g., the effect of a behavioral choice on an individual's health [36]. As an initial motivating example for this work, consider the problem of determining the effect of smoking on an individual's risk of heart disease.

This problem is complicated by the presence of confounders: e.g., it is possible that individuals who smoke have a higher likelihood to be sedentary, which is a lifestyle choice that also negatively impacts their heart disease risk. If the individual's lifestyle is available to us as a well-defined feature, we may adjust for this factor while computing treatment effects [36]. However, confounders are often not observed and not available in the form of features, making accurate causal inference challenging.

Oftentimes, datasets in domains such as medicine or genomics come with large amounts of *unstructured data*—e.g., medical images, clinical notes, wearable sensor measurements [7]. This data contains strong proxy signal about unobserved confounding factors—e.g., wearable sensor measurements can help reveal individuals who are sedentary. However, existing causal inference methods are often not able to leverage this "dark data" for causal effect estimation [18].

The goal of this paper is to develop methods in causal inference that improve the estimation of causal effects in the presence of unobserved confounders by leveraging additional sources of unstructured multi-modal data, such as images and text. For example, given time series from patients' wearables, our methods may disentangle the effects of being sedentary from the effects of smoking by using clusters in the sensor measurements (which would correspond to groups of active and sedentary individuals) as a proxy for the patients' lifestyles and without requiring explicit lifestyle features.

Concretely, our paper formalizes the task of estimating causal effects using rich, unstructured, multi-modal proxy variables (e.g. images, text, time series) and introduces deep multi-modal structural

36th Conference on Neural Information Processing Systems (NeurIPS 2022).

equations, a generative model in which confounders are latent variables. This model can perform causal effect estimation with missing data by leveraging approximate variational inference and learning algorithms [6] developed for this task. Previous methods relied on propensity scoring with neural approximators [41, 50, 51], which may output volatile probabilistic outputs that lead to unreliable effect estimates [18]. Our methods are generative, and thus naturally sidestep some of the aforementioned instabilities, support high-dimensional treatment variables and incomplete data, and as a result are applicable to a broader class of causal inference problems.

We evaluate our methods on an important real-world causal inference task—estimating the effects of genetic mutations in genome-wide association studies (GWASs)—as well as on benchmarks derived from popular causal inference datasets. The intervention variables in a GWAS are high-dimensional genomic sequences, hence existing methods based on propensity scoring are not easily applicable. In contrast, we demonstrate that our algorithms naturally leverage high-dimensional genetic and environmental data (e.g., historical weather time series) and can discover causal genetic factors in plants and humans more accurately than existing GWAS analysis methods.

**Contributions** In summary, this paper makes three contributions: (1) we define the task of estimating causal effects using rich, unstructured multi-modal proxy variables; (2) we introduce deep multi-modal structural equations, a generative model tailored to this problem, and we describe associated variational learning and inference algorithms; (3) we demonstrate on an important real-world problem (GWAS) that unstructured data can improve causal effect estimation, enabling the use of large amounts of "dark data" that were previously not used in causal inference.

## 2 Background

**Notation** Formally, we are given an observational dataset $\mathcal{D} = \{(x^{(i)}, y^{(i)}, t^{(i)})\}_{i=1}^n$ consisting of $n$ individuals, each characterized by features $x^{(i)} \in \mathcal{X} \subseteq \mathbb{R}^d$, a binary treatment $t^{(i)} \in \{0, 1\}$, and a scalar outcome $y^{(i)} \in \mathbb{R}$. We initially assume binary treatments and scalar outcomes, and later discuss how our approach naturally extends beyond this setting. We also use $z^{(i)} \in \mathbb{R}^p$ to model latent confounding factors that influence both the treatment and the outcome [28]. We are interested in recovering the true effect of $T = t$ in terms of its conditional average treatment effect (CATE), also known as the individual treatment effect (ITE) and average treatment effect (ATE).

$$Y[x, t] = \mathbb{E}[Y | X = x, \text{do}(T = t)] \qquad \text{ITE}(x) = Y[x, 1] - Y[x, 0] \qquad \text{ATE} = \mathbb{E}[\text{ITE}(X)], \quad (1)$$

where $\text{do}(\cdot)$ denotes an intervention [37]. Many methods for this task rely on propensity scoring [41, 50, 51], which uses a model $p(t|x)$ to assign weights to individual datapoints; however, when $x$ is high-dimensional and unstructured, a neural approximator for $p(t|x)$ may output volatile and miscalibrated probabilities close to $\{0, 1\}$ that lead to unreliable effect estimates [18].

**Structural Equations** An alternative approach are structural equation models of the form

$$x = f_1(z, \varepsilon_1) \qquad\qquad t = f_2(z, \varepsilon_2) \qquad\qquad y = f_3(z, t, \varepsilon_3), \qquad (2)$$

where $Z \sim p(Z)$ is drawn from a prior and the $\varepsilon_i$ are noise variables drawn independently from their distributions [10]. Structural equations define a *generative model* $p(x, y, z, t)$ of the data. When this model encodes the true dependency structure of the data distribution, we can estimate the true effect of an intervention by clamping $t$ to its desired value and drawing samples.

**Deep Structural Equations** Equations 2 can be parameterized with deep neural networks, which yields deep structural equation models [49, 28]. Expressive neural networks may learn a more accurate model of the true data distribution on large datasets, which improves causal effect estimation. Such models have been used for GWAS analysis [49] and to correct for proxy variables [28].

## 3 Causal Effect Estimation With Unstructured Proxy Variables

Oftentimes, datasets in domains such as medicine or genomics come with large amounts of unstructured data (medical images, clinical notes), which contains strong proxy signal about unobserved confounding factors. Our paper seeks to develop methods that leverage unstructured data within causal inference. We start by formalizing this task as causal effect estimation with unstructured proxy variables; these proxies may come from multiple diverse modalities (images, text).

## 3.1 Task Definition

Formally, consider a causal inference dataset $\mathcal{D} = \{(x^{(i)}, y^{(i)}, t^{(i)})\}_{i=1}^n$ in which $x^{(i)} = (x_1^{(i)}, x_2^{(i)}, \ldots, x_m^{(i)})$ is a vector of $m$ distinct input modalities $x_j^{(i)} \in \mathcal{X}_j$ (e.g., images, text, time series, etc.). In other words, $\mathcal{X} = \mathcal{X}_1 \times \ldots \times \mathcal{X}_m$, where each $\mathcal{X}_j$ corresponds to a space of images, time series, or other unstructured modalities. Here, $t^{(i)} \in \mathcal{T}$ (binary or continuous) is the treatment and $y^{(i)} \in \mathcal{Y}$ is the output. Some modalities may also be missing at training or inference time.

We are interested in recovering the true effect of $t$ in terms of the individual and average treatment effects. We are specifically interested in estimating the individual treatment effect (ITE) from arbitrary subsets of modalities $\mathcal{M} \subseteq \{1, 2, ..., m\}$, indicating that certain inputs may be missing at test time.

$$Y[x, t, \mathcal{M}] = \mathbb{E}[Y | \mathrm{do}(T = t), X_j = x_j \text{ for } j \text{ in } \mathcal{M}] \quad \mathrm{ITE}(x, \mathcal{M}) = Y[x, t = 1, \mathcal{M}] - Y[x, t = 0, \mathcal{M}] \tag{3}$$

To help make this setup more concrete, we define two motivating applications.

**Healthcare** Consider the task of determining the effect of smoking on heart disease from an observational dataset of patients. The observational study may contain additional unstructured data about individuals, e.g., clinician notes, medical images, wearable sensor data, etc. This data may hold information about hidden confounders: for example, raw wearable sensor data can be clustered to uncover sedentary and active indivudals, revealing a latent confounding factor, sedentary lifestyle.

**Genomics** Consider the problem of estimating the effects of genetic variants via a genome-wide association study (GWAS). Modern GWAS datasets in plants or humans feature large amounts of unstructured inputs [7]: clinical notes, medical records, meteorological time series. For example, historical weather data (e.g., precipitation, wind strength, etc.) can reveal distinct climatic regions that affect plant phenotypes and whose confounding effects should be corrected for in a GWAS [55].

# 4 Deep Structural Equations for Causal Effect Estimation

Next, we derive models and inference algorithms for the task of causal effect estimation with unstructured proxy variables. Our approach uses deep structural equations to extract confounding signal from the multi-modal proxies $x_j^{(i)}$. We use neural networks because they naturally handle unstructured modalities via specialized architectures (e.g., convolutions for images) that can learn high-level representations over raw unstructured inputs (e.g., pixels).

Parameterizing structural equations with neural networks also presents challenges: they induce complex latent variable models that require the development of efficient approximate inference algorithms [10, 6]. We present an instantiation of Equations 2 that admits such efficient algorithms.

## 4.1 Deep Multi-Modal Structural Equations

We start by introducing deep multi-modal structural equations (DMSEs), a generative model for estimating causal effects in which confounders are latent variables and unstructured data are proxy variables. We define a DMSE model as follows:

$$z \sim \mathcal{N}(0_p, I_p) \qquad x_j \sim p_{x_j}(\,\cdot\,; \theta_{x_j}(z)) \; \forall j \qquad t \sim \mathrm{Ber}(\pi_t(z)) \qquad y \sim p_y(\,\cdot\,; \theta_y(z, t)), \tag{4}$$

where $p_{x_j}, p_y$ are probability distributions with a tractable density over $x_j$ and $y$, respectively, and the $\theta_{x_j}, \theta_y$ are the parameters of $p_{x_j}, p_y$. The $\theta_{x_j}, \theta_y$ are themselves functions of $z, t$ parameterized by neural networks—e.g., when $x_j$ is Gaussian, the $\theta_{x_j}(z)$ are a mean and a covariance matrix $\mu_{x_j}(z), \Sigma_{x_j}(z)$ that are parameterized by a neural network as a function of $z$ (see [21, 28]). Note that other modeling choices (e.g., Bernoulli distributions for discrete variables) are also possible.

Note that the models for $\theta_{x_j}(z)$ can benefit from domain-specific neural architectures—e.g., a convolutional parameterization for $\mu_{x_j}(z), \sigma_{x_j}(z)$ as a function of $z$ is more appropriate when the $x_j$ are images. See Appendix B for details on recommended architectures.

While Section 3.1 defines $y, t$ as scalars following existing literature [28, 54, 50], DMSEs can also define a model with high-dimensional $y, t$—we simply choose the distributions over $y, t$ to be multi-variate. Our inference and learning algorithms will remain unchanged, except for the parameterization

of specific approximate posteriors (e.g., $q(z|y, t)$). In fact, we apply DMSEs to high-dimensional $t$ in our GWAS experiments in Section 5. Note that this is a setting where existing propensity scoring methods (which learn a model of $p(t|x)$) are not directly applicable [41, 50]—our approach, on the other hand, can easily be used with high-dimensional $t$ on tasks like GWAS analysis.

**Dependency Structure** Equations (4) define a density $p(z)p(t|z)p(y|z, t)\prod_{j=1}^{m} p(x_j|z)$. Note that each proxy $x_j$ is independent of the others conditioned on $z$. Figure 1 shows these dependencies as solid lines. In our setting, the $x_j$ represent image pixels, waveform measurements, etc; thus, they should not directly influence each other or $y, t$. For example, it would not make sense for the pixels of an image to causally influence the samples of a waveform measurement—they influence each other only through a latent confounder $z$ (e.g., patient health status), and the $x_j$ are therefore assumed to be conditionally independent given $z$. These independence assumptions will also enable us to derive efficient stochastic variational inference algorithms, as we show below.

In certain settings, we may also want to model observed confounders as well as additional *structured* proxies $x_j$ that have direct causal effects on each other and on $y, t$. Figure 1 shows these additional dependencies as dashed lines. Our model admits simple generalizations that support these modeling assumptions. In brief, our variational inference algorithms can be conditioned on the observed variables, and mutually dependent sets of proxies can be treated as a group represented by one high-dimensional variable. See Appendix H for details.

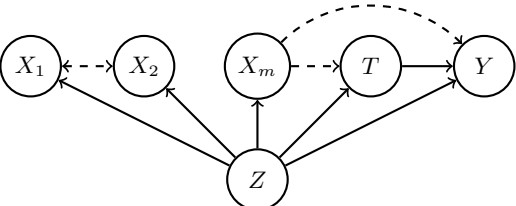

Figure 1: Causal graph for the DMSE model. Solid lines depict dependencies between variables. Appendix H contains simple extensions of DMSEs that support extra dependencies (dashed lines).

### 4.2 Approximate Inference and Learning Algorithms

In the full multi-modal setting, the $x_j$ are conditionally independent given $z$ (Figure 1), which enables us to apply efficient algorithms inspired by Wu & Goodman [56]. These algorithms offer the following improvements: (1) we may perform learning, inference, and causal estimation with missing modalities $x_j$; (2) the process for performing causal inference does not require training auxiliary inference networks as in previous work [28].

The DMSE model induces a tractable joint density $p(x, y, t, z)$, which allows us to fit its parameters using stochastic variational inference by optimizing the evidence lower bound (ELBO):

$$\text{ELBO}_X = \sum_{i=1}^{n} \mathbb{E}_q \Big[ \sum_{j=1}^{m} \log p(x_j^{(i)}|z) + \log p(y^{(i)}, t^{(i)}, z) - \log q(z|x^{(i)}, y^{(i)}, t^{(i)}) \Big], \quad (5)$$

where $p(y^{(i)}, t^{(i)}, z) = p(y^{(i)}|t^{(i)}, z)p(t^{(i)}|z)p(z)$ and $q(z|x^{(i)}, y^{(i)}, t^{(i)})$ is the approximate variational posterior. We assume a total of $m$ modalities.

**Structured Multi-Modal Variational Inference** We may use the independence structure of $p$ (Figure 1) to derive an efficient structured form for $q$. First, observe that because the true posterior factorizes as $p(z|x, t, y) \propto (p(z|t, y)\prod_{j=1}^{m} p(z|x_j))/\prod_{j=1}^{m-1} p(z)$, the optimal approximate posterior $q$ must also factorize as $q(z|x, t, y) \propto (q(z|t, y)\prod_{j=1}^{m} q(z|x_j))/\prod_{j=1}^{m-1} p(z)$. This decomposition implies that we can maintain the optimal structure of $q$ by training modality-specific inference networks $\tilde{q}(z|t, y)$ and $\tilde{q}(z|x_j)$ such that $q(z|x_j) = \tilde{q}(z|x_j)p(z)$ and $q(z|t, y) = \tilde{q}(z|t, y)p(z)$ and by defining a joint posterior as

$$q(z|x, y, t) \propto p(z)\tilde{q}(z|y, t)\prod_{j=1}^{m} \tilde{q}(z|x_j). \quad (6)$$

This network can be seen as a product of experts (PoE) [56].

Computing the density $q$ is in general not possible. However, because $p(z)$, $q(z|t, y)$ and $q(z|x_j)$ are Gaussians, we may use the fact that a product of Gaussians with means $\mu_i$ and covariances $V_i$ is $\mu = (\sum \mu_i T_i)/(\sum T_i)$ and $V = (\sum_i T_i)^{-1}$, where $T_i = 1/V_i$. Thus, computing $q(z|x, t, y)$ for

any subset of modalities is possible without having to train an inference network for each subset of modalities separately.

**A Multi-Modal ELBO**   Training this architecture with $\text{ELBO}_X$ will not necessarily yield good single-modality inference networks $\tilde{q}(z|x_j)$, while training with each modality separately will prevent the network from learning how the modalities are related to each other. Hence, we train our model with a sub-sampled ELBO objective [56] that is computed on the full set of modalities, each individual modality, and a few subsets of modalities together. For this, we randomly pick at each gradient step $s$ non-empty subsets $\{\mathcal{M}_k\}_{k=1}^s$ of the set of modalities $\mathcal{M}_k \subseteq \{1, 2, ..., m\}$. The final objective is $\text{ELBO}_X + \sum_{j=1}^m \text{ELBO}_{\{j\}} + \sum_{k=1}^s \text{ELBO}_{\mathcal{M}_k}$, where

$$\text{ELBO}_{\mathcal{M}} = \sum_{i=1}^n \mathbb{E}_q \left[ \sum_{j \in \mathcal{M}} \log p(x_j^{(i)}, z) + \log p(y^{(i)}, t^{(i)}, z) - \log(p(z)\tilde{q}(z|y^{(i)}, t^{(i)}) \prod_{j \in \mathcal{M}} \tilde{q}(z|x_j^{(i)})) \right]$$

Our model doesn't train extra auxiliary inference networks, unlike that of Louizos et al. [28].

### 4.3   Deep Gaussian Structural Equations

The DMSE model can be simplified in settings in which there is only one type of proxy $x$ (i.e., $m = 1$). This simplified model, which we call deep Gaussian structural equations (DGSEs), has a tractable joint density $p(x, y, t, z) = p(z)p(x|z)p(t|z)p(y|z, t)$, where the latent $z$ is Gaussian.

The DGSE model can also be fit using stochastic variational inference by optimizing the ELBO objective $\sum_{i=1}^n \mathbb{E}_q \left[ \log p(x^{(i)}, y^{(i)}, t^{(i)}, z) - \log q(z|x^{(i)}, y^{(i)}, t^{(i)}) \right]$, where $q(z|x, y, t)$ is an approximate variational posterior. We optimize the above objective using gradient descent, applying the reparameterization trick to estimate the gradient. We compute the counterfactual $Y[x, t]$ using auxiliary inference networks as in earlier work [28]. See Appendix B for the full derivation.

### 4.4   Properties of Deep Multi-Modal Structural Equations

**Recovering Causal Effects**   The DMSE and DGSE models determine the true causal effect when their causal graph is correct and they recover the true data distribution. The following argument is analogous to that made in most previous works on causal deep generative models [28, 57, 35].

**Theorem 1.** *The DMSE and DGSE models recover the true ITE$(x, \mathcal{M})$ for any subset $\mathcal{M} \subseteq \{1, 2, ..., m\}$ of observed modalities whenever they represent the true data distribution $p(x, y, t, z)$.*

**Proof:** We establish the theorem for DMSEs; the proof for DGSEs is analogous with $m = 1$. Let $x_{\mathcal{M}} = \{x_j \mid j \in \mathcal{M}\}$ be the data from the observed subset of modalities. We need to show that $p(y|x_{\mathcal{M}}, \text{do}(t = t'))$ is identifiable for any $t'$. Observe that

$$p(y|x_{\mathcal{M}}, \text{do}(t = t')) = \int_z p(y|z, x_{\mathcal{M}}, do(t = t'))p(z|x_{\mathcal{M}}, do(t = t'))dz = \int_z p(y|z, x_{\mathcal{M}}, t')p(z|x_{\mathcal{M}})dz,$$

where the second equality follows from the rule of do-calculus (applying backdoor adjustment). Since our proof holds for any $t'$ and all elements on the right-hand side are identifiable, the claim follows. ∎

Note that in practice our assumption may not hold (e.g., neural network optimization is non-convex and may fail), but there is evidence of both failure modes [44] as well as successful settings in which deep latent variable models provide useful causal estimates [57, 35, 30, 48]. See our Discussion section for additional details.

**Identifiability in Linear Models**   Structural equations parameterized by non-convex neural networks are less amenable to analysis than simpler model classes. However, we may provide theoretical guarantees in the special case where a DMSE model (Equations 4) is linear, i.e., each equation with input variables $u \in \mathbb{R}^{d_1}$ has the form $A \cdot u + b$ for some $A \in \mathbb{R}^{d_2 \times d_1}, b \in \mathbb{R}_1^d$. Specifically, we establish in Appendix I the following result.

**Theorem 2.** *Given a binary treatment $t$, a univariate outcome $y$, confounder $z$ and proxy variables $u, v, w$, the causal effect $P(y|do(t))$ is identifiable if*

1. *The structural equations follow a DMSE model (Figure 1, solid edges) and are linear.*

2. *Three independent views of $z$ are available in the form of proxies $u, v, w$ such that $u \perp v \perp w|z$ and the equations between $z$ and $u, v, w$ are parameterized by matrices of rank $dim(z)$*

Our proof extends techniques developed by Kuroki and Pearl [22] to high-dimensional proxy variables. Interestingly, our result crucially relies on an independence structure specified by Figure 1 (specifically, the existence of three independent proxy variables), which lends additional support for this modeling assumption and for the development of variational techniques specialized to this model family.

**Computing Causal Effects** Given a subset of modalities $\mathcal{M}$, we can compute the ATE & ITE as $\mathbb{E}(\text{ITE}(x, \mathcal{M}))$, where $\text{ITE}(x, \mathcal{M}) = Y[x, t = 1, \mathcal{M}] - Y[x, t = 0, \mathcal{M}]$ and

$$p(y|x, \text{do}(t = t')) = \int_z p(y|t = t', z)p(z|x)dz \approx \int_z \big(p(y|t = t', z)p(z) \prod_{j=1}^m \tilde{q}(z|x_j)\big)dz, \quad (7)$$

where we use our variational posterior formulation from Equation 6 to approximate the true posterior $p(z|x)$.

## 5 Experimental Results

### 5.1 Synthetic Demonstration Dataset

We start with a demonstration that provides intuition for why proxy variables are important, and how unstructured proxies can serve in place of featurized (structured) ones. The following small-scale synthetic setup (Louizos et al. [28]) involves a data distribution $\mathbb{P}$ over binary variables $y, t, z, x$:

$$\mathbb{P}(z = 1) = \mathbb{P}(z = 0) = 0.5 \quad \mathbb{P}(x = 1|z = 1) = \rho_{x1} = 0.3 \quad \mathbb{P}(x = 1|z = 0) = \rho_{x0} = 0.1$$
$$y = t \oplus z \quad \mathbb{P}(t = 1|z = 1) = \rho_{t1} = 0.4 \quad \mathbb{P}(t = 1|z = 0) = \rho_{t0} = 0.2$$

where $0 < \rho_{x1}, \rho_{x0}, \rho_{t1}, \rho_{t0} < 1$ are parameters. We also introduce an unstructured proxy variable $\mathbf{X}$ that represents an "image version" of $x$. The variable $\mathbf{X}$ will be a random MNIST image of a zero or one, depending on whether $x = 0$ or $x = 1$. Formally, $\mathbf{X}$ is distributed as follows:

$$\mathbb{P}(\mathbf{X}|x = 1) \text{ is unif. over MNIST images of '1'} \quad \mathbb{P}(\mathbf{X}|x = 0) \text{ is unif. over MNIST images of '0'}$$

Table 1: Treatment effect estimation on the synthetic demonstration dataset.

| | Setting | $\varepsilon_{ATE}$ (Train) | $\varepsilon_{ATE}$ (Test) |
|---|---|---|---|
| Deep Str Eqns | Binary | 0.062 (0.012) | 0.069 (0.015) |
| | Image | 0.068 (0.018) | 0.096 (0.018) |
| IPTW [29] | Binary | 0.090 (0.005) | 0.127 (0.016) |
| | Image | 5.050 (0.607) | 4.067 ( 0.533) |
| Augmented | Binary | 0.442 (0.040) | 0.487 (0.037) |
| IPTW [45] | Image | 4.717 (0.670) | 6.426 (1.603) |
| Non-Causal | Binary | 0.197 (0.003) | 0.206 (0.004) |
| Baseline | Image | 0.214 ( 0.026) | 0.228 ( 0.025) |

First, this is a setup that requires us to model proxies : treating $\mathbf{X}$ as a confounder as using a model of $\mathbb{P}(y \mid \mathbf{X}, t)$ recovers the true ATE only when $\rho_{t1} = 1 - \rho_{t0}$ and $\rho_{x1} = 1 - \rho_{x0}$ (i.e., when $\mathbf{X}$ is perfectly informative of $z$), otherwise it fails (see also Appendix A).

We also show that structural equations solve this task. We sample 3000 data points from $\mathbb{P}$ and fit DGSE models to 80% of the data points $\{x, y, t\}$ (the BI-NARY setting) as well as on $\{\mathbf{X}, y, t\}$ (the IMAGE setting). We note the Average Treatment Effect (ATE) on the training and test sets, and we report results in Table 1. We compare DGSE with the Inverse Probability of Treatment Weighted estimator (IPTW) [29] and the doubly robust Augmented-IPTW [45]—in each case the propensity score model is an MLP trained to predict $t$ from either $x$ or $\mathbf{X}$. We found that replacing $x$ with an image $\mathbf{X}$ causes the model to output highly miscalibrated probabilities close to $0, 1$ (while maintaining good accuracy), which results in large and volatile inverse propensity weights and in poor ATE estimates.

## 5.2 Benchmark Datasets for Causal Effect Estimation

**IHDP** The Infant Health and Development Project (IHDP) is a popular benchmark for causal inference algorithms [13] that contains the outcomes of comprehensive early interventions for premature, low birth weight infants. We create a benchmark for multi-modal causal inference based on IHDP in which we can replace certain features with their "unstructured version". We choose 9 of the 25 features available in IHDP in order to magnify their relative importance and accurately measure the effects of their removal. Please refer to Appendix D and E for detailed setup.

**STAR** The Student-Teacher Achievement Ratio (STAR) experiment [1] studied the effect of class size on the performance of students. We consider small class size as treatment; the outcome is the sum of the reading and math scores of a student. We 'derandomize' this dataset by removing 80% of the data corresponding to white students in the treated population. Similarly to IHDP, we select 8 attributes for the multi-modal experiment. Further details can be found in the Appendix D and E.

**Adding Unstructured Modalities** We create a benchmark for multi-modal causal inference derived from IHDP and STAR in which we replace features with unstructured inputs that contain the same information as their featurized versions. On IHDP, we replace the attribute 'baby's gender' with the CLIP embedding [24] of an image of a child between ages 3 to 8 years, drawn from the UTK dataset [61]. On STAR, we replace the attributes corresponding to the student's ethnicity and gender by selecting an image of a child with the same ethnicity and gender from the UTK dataset.

Table 2: Multimodal Experiments on IHDP Dataset: With deep structural equations, replacing baby's gender with corresponding image embedding (8 attrs + image) shows some increase in ATE error as compared to IHDP-Mini setting (9 attrs) but is better than dropping this modality altogether (8 attrs).

| Model | $\varepsilon_{ATE}$ (Train+Val) | $\varepsilon_{ATE}$ error(Test) |
|---|---|---|
| Deep Str Eqns | | |
| 9 attrs | 0.259 (0.037) | 0.487 (0.078) |
| 8 attrs | 0.392 (0.141) | 0.620 (0.158) |
| 8 attrs + image | 0.372 (0.107) | 0.575 (0.130) |
| CFRNet | | |
| 9 attrs | 0.433 (0.063) | 0.549 (0.090) |
| 8 attrs | 0.412 (0.062) | 0.608 (0.107) |
| 8 attrs + image | 0.501 (0.076) | 0.617 (0.114) |
| OLS | | |
| 9 attrs | 0.424 (0.061) | 0.584 (0.100) |
| 8 attrs | 0.429 (0.066) | 0.593 (0.103) |
| 8 attrs + image | 0.428 (0.064) | 0.590 (0.101) |

We train and evaluate models on datasets where the image 'replaces' the attribute. (e.g., 8 ATTRS + IMAGE). We also consider two other settings for comparison: a) the original attribute is included (e.g., 9 ATTRS) and b) the attribute is dropped from the reduced set of input features (8 ATTRS).

**Results** As seen in Table 2, the degradation in ATE error from replacing the baby's gender by a photograph is lower as compared to removing the attribute entirely. This shows our models leverage signal found in the unstructured image modality with the help of deep neural networks.

We compare these results with a simple Ordinary Least Squares model (OLS) baseline as described by Shalit et al. [47] to predict treatment effect. OLS shows a similar behavior when replacing baby's gender with corresponding image, however ATE errors are generally worse as compared to ATE error produced by DGSE. We also compare this with Counterfactual Regression Network (CFRNet) [16] baseline. However CFRNet did not show benefits of using image modality unlike our approach.

In Table 6, replacing gender and ethnicity attributes on STAR with the corresponding image improves ATE errors as compared to dropping these two attributes entirely. This shows that we can use an image to extract multiple attributes while doing causal inference. The CFRNet baseline shows a similar behavior, but the difference between average ATE errors across different setups is small.

## 5.3 Genome-Wide Association Studies

We evaluate our methods on an important real-world causal inference problem—genome-wide association study analysis (GWASs). A GWAS is a large observational study that seeks to determine the causal effects of genetic markers (or genotypes) on specific traits (known as phenotypes). In this setting, treatment variables are high-dimensional genomic sequences, and existing propensity scoring methods (which learn a model of $p(t|x)$ for a binary $t$) are not easily applicable [41, 50]—they may

require training an impractical number of models. Our approach, on the other hand, can easily be used with high-dimensional $t$.

**Background and Notation** As motivation, consider the problem of linking a plant's genetic variants $t \in \{0,1\}^d$ with *nutritional yield*, which we model via a variable $y \in \mathbb{R}$. Our goal is to determine if each variant $t_j$ is *causal* for yield, meaning that it influences biological mechanisms which affect this phenotype [8]. We also want to leverage large amounts of unstructured data $x$ (e.g., health records, physiological data) that are often available in modern datasets [7, 25].

A key challenge in finding causal variants is ancestry-based confounding [2, 52]. Suppose that we are doing a GWAS of plants from Countries A and B; plants in Country A get more rainfall, and thus grow faster and are more nutritious. A simple linear model of $y$ and $t$ will find that any variant that is characteristic of plants in Country A (e.g., bigger leaves to capture rain) is causal for nutritional yield.

Table 3: Comparison of standard DSE methods with linear baselines. The $\ell_1$ column refers to $\|\widehat{\gamma} - \gamma^\star\|_1$ where $\widehat{\gamma}$ is the vector of estimated causal effects, and $\gamma^\star$ is the vector of ground truth causal effects. Precision and recall are defined in Appendix G. Standard error of the Mean (sem) is computed over 10 seeds.

| Model | $\ell_1 (\downarrow)$ Mean (sem) | Precision ($\uparrow$) Mean (sem) | Recall ($\uparrow$) Mean (sem) |
|---|---|---|---|
| Optimal | 0.22 (0.04) | 0.97 (0.03) | 1.0 (0.00) |
| DSE (2 modalities) | 0.30 (0.06) | 0.93 (0.04) | 1.0 (0.00) |
| LMM | 0.44 (0.06) | 0.85 (0.08) | 1.0 (0.00) |
| DSE (1 modality) | 0.60 (0.09) | 0.78 (0.08) | 1.0 (0.00) |
| PCA (1 component) | 0.93 (0.17) | 0.58 (0.09) | 1.0 (0.00) |
| FA (1 component) | 1.08 (0.17) | 0.62 (0.08) | 1.0 (0.00) |
| PCA (2 components) | 1.38 (0.24) | 0.44 (0.09) | 0.9 (0.07) |
| FA (2 components) | 1.44 (0.30) | 0.55 (0.09) | 1.0 (0.00) |
| PCA (3 components) | 1.66 (0.23) | 0.37 (0.08) | 0.8 (0.08) |
| FA (3 components) | 1.89 (0.45) | 0.44 (0.08) | 0.9 (0.07) |

**Methods and Baselines** Most existing GWAS analysis methods for estimating the effect of a variant $t_j$ rely on latent variable models: (1) they treat all remaining variants $x$ as proxies and obtain $z$ via *a linear projection* (e.g., PCA [38, 39] or LMM [59, 26]) of $x$ into a lower dimensional space where genomes from Countries A and B tend to form distinct clusters (because plants from the same country breed and are similar); (2) we assume a *linear model* $\beta^\top t$ of $y$ and add $z$ into it, which effectively adds the country as a feature ($z$ reveals the cluster for each country); this allows the model to *regress out* the effects of ancestry and assign the correct effect to variants $t$ (one at a time).

*Baselines.* Our main baselines are Principal Component Analysis (PCA) and Linear Mixed Model (LMM), as described above and implemented via the popular LIMIX library [27]. We also compare against Factor Analysis (FA), a standard linear technique for deriving latent variables, Uniform Manifold Approximation and Projection (UMAP), a manifold learning technique for dimensionality reduction [31] and a linear model with no correction for confounding.

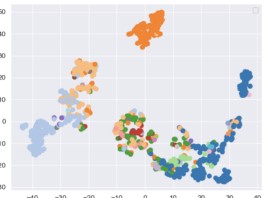

*DSE Models.* We compare against deep structural equation models that leverage one or more sets of proxy variables coming from the following unstructured modalities: genomic sequences, weather time series, simulated physiological time series. We fit DSE models via a stagewise strategy analogous to how classical GWAS models are

Figure 2: Latent $z$ extracted by DSEs. Plants from different countries form clusters (in color).

fit: (1) we fit the component $p(z_k|x_k)$ for each proxy $x_k$; (2) we fit a *linear model* of $y$ given $t$ and the $z$ to estimate causal effects. Thus, our $p(x_k|z)$ components are deep, while $p(y|x,z)$ is shallow (following standard assumptions on epistasis in GWAS).

### 5.3.1 Simulated Human GWAS

We used the 1000 Human Genomes [3] dataset to generate a simulated multi-modal GWAS dataset, following the biologically-inspired "Spatial" strategy studied in [49]. In addition to genotypes, we generated random physiological time series by sampling Fourier series conditioned on the confounders. Please see Appendix G for details.

Table 3 shows our results. Amongst the non-oracle baselines, multi-modal DSE has the smallest error in estimating causal effect, as well as the highest precision and recall at identifying causal SNPs. The uni-modal DSE, while worse than LMM, still beats PCA and FA. Note that the LMM

model is not compatible with multiple unstructured proxies. In general, we see that precision starts to deteriorate faster than recall, suggesting that false positives are more likely from the weaker linear deconfounding methods such as PCA/FA. Our results here further support that additional sources of unstructured multi-modal data can improve GWAS.

### 5.3.2 Real-World Genomic Prediction and GWAS in Plants

We also tested our methods on a real-world plant GWAS dataset from the 1001 Genomes Project for *Arabidopsis Thaliana* plants [19, 55]. The challenge here is that we don't know true causal effects; therefore, we define a phenotype $y$ for which the true causal effect is zero—specifically, we set $y$ to the GDP of the country where each plant grows. Because the genomes of plants from the same country are similar, there exist spurious correlations caused by latent subpopulation groups $z$. Our goal is to detect and correct for these con-

Table 4: Multimodal Experiments on IHDP and STAR Datasets comparing DMSE and DGSE methods. When the image modality is missing, DMSE still produces reasonable ATE errors as it can compute the latent representation on a subset of modalities better as compared to DMSE

| Dataset | Model | $\varepsilon_{ATE}$ (train+val) | $\varepsilon_{ATE}$ error(test) |
|---------|-------|------------------|------------------|
| IHDP | DMSE | 0.433 (0.057) | 0.627 (0.094) |
|      | DGSE | 0.794 (0.308) | 1.080 (0.337) |
| STAR | DMSE | 32.575 (1.634) | 33.743 (1.890) |
|      | DGSE | 59.102 (4.734) | 60.152 (4.788) |

founding effects. In addition to genomic data, we gather historical weather time series from the location of each plant (see Appendix F) and use both modalities to correct for confounding.

Table 5: Correcting for confounding on a plant GWAS dataset. DSEs can discover complex, non-linear clusters over genomic and weather data to identify the latent confounding variable better.

| Model | Input | $R^2(\downarrow)$ |
|-------|-------|-------------------|
| Deep Str Eqns (Ours) | Weather+SNPs | 0.049 (0.022) |
| PCA | Weather+SNPs | 0.097 (0.019) |
| UMAP | Weather+SNPs | 0.412 (0.026) |
| Deep Str Eqns (Ours) | Weather | 0.555 (0.027) |
| PCA | Weather | 0.545 (0.017) |
| UMAP | Weather | 0.653 (0.013) |
| Deep Str Eqns (Ours) | SNPs | 0.068 (0.029) |
| PCA | SNPs | 0.130 (0.020) |
| UMAP | SNPs | 0.406 (0.020) |
| LMM | SNPs | 0.804 (0.009) |
| Linear Model | - | 0.665 (0.012) |

We evaluate whether each model learned to correctly account for latent confounding effects by measuring the predictive power of $\beta^\top t$ for $y$, where $\beta$ are the causal effects and $t$ is the vector of variants. Specifically, we compare the $R^2$ correlation between $\beta^\top t$ and $y$—here, *lower is better*, since a model that has learned the causal effects should not be predictive of the phenotype. In Table 5, we see the effect of extracting confounding variables using DSEs as opposed to using the standard PCA technique. We can see that the $R^2$ values produced using DSEs are closer to $0$ as compared to using PCA. This experiment shows that neural network architectures are effective in dealing with unstructured genomic and weather data while correcting for confounding. Please refer to Appendix F for details.

### 5.4 Multimodal Experiments With Missing Modalities

We demonstrate the ability of the DMSE model to handle missing data. Our IHDP and STAR benchmarks involve two modalities: images (e.g., baby's gender in IHDP) and tabular data (e.g., the remaining features). We compare DMSE and DGSE models on these datasets when some of the modalities may be missing.

For DMSE, we define two different modalities $X_1$ and $X_2$ for the tabular and image modalities respectively. For DGSE, we concatenate the image embedding to the tabular modality while training the network. We evaluate ATE while randomly dropping 50% of the images. DMSE utilizes its product-of-experts inference network to approximate the posterior distribution when modalities are missing. DGSE cannot do this, and we resort to feeding it a vector of zeros when an image is missing. Table 4 shows that DMSE produces improved ATE estimates as compared to DGSE.

# 6 Related Work

**Multi-Modal Causal Inference** While previous work analyzed unstructured interventions $t$ consisting of natural language [41–43] (e.g., determining the effect of a polite vs. a rude response) as well as unstructured $y$ [5, 35] (e.g., MRI images), our work proposes methods to handle unstructured $x$. Veitch et al. [50] developed models that correct for confounders from a single unstructured proxy $x$ derived from text [33, 51] or a graph [50]. These approaches rely on a propensity scoring framework—they train a discriminative model of $p(t|x)$—hence do not support proxy variables or missing data, and require pre-trained text embeddings. Additionally, propensity scoring methods rely on neural approximators for $p(t|x)$, which may output volatile probabilistic outputs that lead to unreliable effect estimates [18]. Our method (i) works across all modalities (beyond text or graphs), (ii) supports arbitrary numbers of proxies, (iii) supports missing data by virtue of being generative.

**Deep Latent Variable Models** Representation learning in causal inference has been studied by Johansson et al. [16, 15, 17] and Schölkopf et al. [46]. Deep latent variable models find applications throughout causal inference [28, 30, 57, 35, 60, 53, 20]. Pawlowski et al. [35] study unstructured outcomes $y$ (MRI scans), but do not support proxies. Louizos et al. [28] use variational auto-encoders to estimate confounders from proxies; we introduce a more structured model that handles multiple proxies that can be missing, and obviates the need for auxiliary modules. Tran & Blei [49] propose implicit deep structural equations for GWAS; ours are explicit and thus easier to train.

# 7 Discussion

**Identifiability** Approaches to causal effect estimation that rely on deep learning [28, 47, 35, 48, 30, 11, 60, 58] can never guarantee the recovery of causal effects—neural network optimization is itself non-convex and has no guarantees. Other failure modes of deep latent variable models (DLVMs) include potentially not having a sufficiently expressive model, not having enough data to learn the model, as well as shortcomings of approximate inference algorithms. That said, there is ample evidence of both failures [44] and successes of DLVMs [57, 35, 30]. The DLVM approach is appealing over existing propensity scoring methods [41, 50] because: (i) it naturally handles unstructured proxies that may be missing at random; (ii) it supports high-dimensional treatment variables in settings like GWAS, where propensity scoring algorithms are not easily applicable. Rissanen & Marttinen [44] empirically identify multiple failure modes of DLVMs; our work and that of others identifies success cases (particularly in GWAS [49, 54]), and ultimately the validation of DLVM methods needs to be empirical [48, 58, 50, 11, 20].

**Missing Data** We make the common assumption that data is missing at random (MAR). This poses challenges if, for example, patients missing outcomes are ones that are more likely to be sick. When two modalities and their missingness are correlated, their $x_j, x_k$ nodes in Figure 1 could be merged, somewhat addessing the issue. We leave the full exploration of non-MAR models to future work.

# 8 Conclusion

In conclusion, we proposed an approach based on deep structural equations that can leverage useful signal present in unstructured data to improve the accuracy of causal effect estimation. Unlike previous methods that relied on propensity scores [41, 50], ours does not suffer from instabilities caused by volatile predictive probabilities coming out of neural networks, naturally handle missing data, and are applicable in settings in which the treatment variable is high-dimensional (such as in genome-wide association studies). Our work highlights the benefits of using large amounts of "dark" data that were previously left unused by existing methods to improve the accuracy of causal effect estimation.

# Acknowledgements

This work was supported by Tata Consulting Services, the Cornell Initiative for Digital Agriculture, and an NSF CAREER grant (#2145577).

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
