# A  Details of the toy experiment.

1. **Dataset.** In this experiment, we generate synthetic dataset consisting of 5 variables: a latent binary variable $Z$, a binary variable $X^*$, an MNIST image $\mathbf{X}$, and two other binary variables $y, t$ standing for outcome and treatment respectively.

$$
\begin{aligned}
&Z_i, X_i^* \in \{0, 1\} \\
&\mathbf{X}_i \in \mathbb{R}^{28*28} \quad \text{is an MNIST image of 0 or 1} \\
&y_i, t_i \in \{0, 1\} \\
&i \in \{1, 2, \cdots, N\}, N = 3000
\end{aligned}
$$

These variables are sampled from the following distribution:

$$
\begin{aligned}
&P[Z = 0] = P[Z = 1] = 0.5 \\
&P[t = 1|Z = 0] = \rho_{t0}, P[t = 1|Z = 1] = \rho_{t1} \\
&P[X^* = 1|Z = 0] = \rho_{x0}, P[X^* = 1|Z = 1] = \rho_{x1} \\
&\mathbf{X} = \begin{cases} \text{Binarized image of 0 when } X^* = 0 \\ \text{Binarized image of 1 when } X^* = 1 \end{cases} \\
&y = t \oplus Z
\end{aligned}
$$

2. **Accounting for Confounding in ATE Computation.**

As mentioned before, the definition of the Average Treatment Effect (ATE) is as follows:

$$
\begin{aligned}
ATE &= \mathbb{E}[ITE(x)] \\
&= \mathbb{E}\Big[\mathbb{E}[y \mid \mathbf{X} = x, do(t = 1)] - \mathbb{E}[y \mid \mathbf{X} = x, do(t = 0)]\Big]
\end{aligned}
$$

Now we consider the first term within the outer expectation:

$$
\begin{aligned}
&\mathbb{E}[\, y \mid \mathbf{X} = x, \ do(t = 1)] \\
&= 1 * P[y = 1|\mathbf{X} = x, do(t = 1)] + 0 \\
&= \sum_z P[y = 1|\mathbf{X} = x, do(t = 1), Z = z] \cdot P[Z = z|\mathbf{X} = x, do(t = 1)] \\
&= \sum_z P[y = 1|\mathbf{X} = x, t = 1, Z = z] \cdot P[Z = z|\mathbf{X} = x] \\
&= P[y = 1|\mathbf{X} = x, t = 1, Z = 0] \cdot P[Z = 0|\mathbf{X} = x] + 0 \\
&= P[Z = 0|\mathbf{X} = x]
\end{aligned}
$$

where the second last equation is due to the fact that $y = t \oplus Z$. Similarly, we can compute the second term as

$$
\mathbb{E}[\, y \mid \mathbf{X} = x, do(t = 0)] = P[Z = 1|\mathbf{X} = x].
$$

Using the predefined generative process for this dataset, we can also write

$$
\begin{aligned}
P[Z = z|\mathbf{X} = x] &= \frac{P[Z = z] \cdot P[\mathbf{X} = x|Z = z]}{P[\mathbf{X} = x]} \\
&= \frac{0.5 \cdot P[\mathbf{X} = x|Z = z]}{P[\mathbf{X} = x]} \\
&= \frac{0.5 \cdot P[X^* = x^*|Z = z]}{P[\mathbf{X} = x]}
\end{aligned}
$$

where $x$ is an image of binary variable $x^*$. Plugging in the previous results, we can compute the individual treatment effect (ITE):

$$
\begin{aligned}
& ITE(x) \\
= {} & \mathbb{E}[y|\mathbf{X} = x, do(t = 1)] - \mathbb{E}[y|\mathbf{X} = x, do(t = 0)] \\
= {} & \frac{P[Z = 0|\mathbf{X} = x] - P[Z = 1|\mathbf{X} = x]}{P[\mathbf{X} = x]} \\
= {} & \frac{P[\mathbf{X} = x|Z = 0] - P[\mathbf{X} = x|Z = 1]}{P[\mathbf{X} = x]} \\
= {} & \frac{P[X^* = x^*|Z = 0] - P[X^* = x^*|Z = 1]}{P[\mathbf{X} = x]} \\
= {} & \begin{cases} 0.5 \cdot (\rho_{x1} - \rho_{x0})/(P[X^* = 0]), & \text{if } x = \text{image of } 0 \\ 0.5 \cdot (\rho_{x0} - \rho_{x1})/(P[X^* = 1]), & \text{if } x = \text{image of } 1 \end{cases}
\end{aligned}
$$

Therefore we can plug in the previous equation and get the final result of ATE:

$$
\begin{aligned}
ATE = {} & \mathbb{E}[ITE(x)] \\
= {} & \sum_x P[\mathbf{X} = x] \cdot ITE(x) \\
= {} & \sum_{x^*} P[X^* = x^*] \cdot ITE(\text{image of } x^*) \\
= {} & 0.5 \cdot ((\rho_{x0} - \rho_{x1}) + (\rho_{x1} - \rho_{x0})) \\
= {} & 0
\end{aligned}
$$

3. **ATE Computation with Non-Causal Model.** ATE computation goes wrong when $\mathbf{X}$ is taken to be the only confounder. In the following computation of $\mathbb{E}[\, y \mid \mathbf{X} = x, \, do(t = 1)]$, the second step goes wrong as adjustment is done over $\mathbf{X}$ instead of $Z$.

$$
\begin{aligned}
& \mathbb{E}[\, y \mid \mathbf{X} = x, \, do(t = 1)] \\
= {} & 1 * P[y = 1|\mathbf{X} = x, do(t = 1)] + 0 \\
= {} & \sum_x P[y = 1|\mathbf{X} = x, t = 1] \cdot P[\mathbf{X} = x] \\
= {} & P(\mathbf{X} = \text{image of } 1) \cdot P[y = 1|\mathbf{X} = \text{image of } 1, t = 1] + \\
& P(\mathbf{X} = \text{image of } 0) \cdot P[y = 1|\mathbf{X} = \text{image of } 0, t = 1] \\
= {} & P(X^* = 1) \cdot P[y = 1|X^* = 1, t = 1] + P(X^* = 0) \cdot P[y = 1|X^* = 0, t = 1] \\
= {} & P(X^* = 1) \cdot \frac{P[y = 1, X^* = 1, t = 1]}{P[X^* = 1, t = 1]} + P(X^* = 0) \cdot \frac{P[y = 1, X^* = 0, t = 1]}{P[X^* = 0, t = 1]} \\
= {} & P(X^* = 1) \cdot \frac{\sum_z (P[y = 1, X^* = 1, t = 1|Z = z]P(Z = z))}{\sum_z (P[X^* = 1, t = 1|Z = z]P(Z = z))} \\
& + P(X^* = 0) \cdot \frac{\sum_z (P[y = 1, X^* = 0, t = 1|Z = z]P(Z = z))}{\sum_z (P[X^* = 0, t = 1|Z = z]P(Z = z))} \\
= {} & P(X^* = 1) \cdot \frac{(P[y = 1, X^* = 1, t = 1|Z = 0] \cdot 0.5)}{\sum_z (P[X^* = 1, t = 1|Z = z] \cdot 0.5)} \\
& + P(X^* = 0) \cdot \frac{(P[y = 1, X^* = 0, t = 1|Z = 0] \cdot 0.5)}{\sum_z (P[X^* = 0, t = 1|Z = z] \cdot 0.5)} \\
= {} & P(X^* = 1) \cdot \frac{\rho_{x0}\rho_{t0}}{\rho_{x1}\rho_{t1} + \rho_{x0}\rho_{t0}} + P(X^* = 0) \cdot \frac{((1 - \rho_{x0})\rho_{t0})}{(1 - \rho_{x1})\rho_{t1} + (1 - \rho_{x0})\rho_{t0}}
\end{aligned}
$$

Similarly, we get wrong expectation when performing intervention on variable $t$ to set it to zero.

$$\mathbb{E}[\, y \mid \mathbf{X} = x, \; do(t = 0)] \; = P(X^* = 1) \cdot \frac{\rho_{x1}(1 - \rho_{t1})}{\rho_{x1}(1 - \rho_{t1}) + \rho_{x0}(1 - \rho_{t0})} +$$

$$P(X^* = 0) \cdot \frac{((1 - \rho_{x1})(1 - \rho_{t1}))}{(1 - \rho_{x1})(1 - \rho_{t1}) + (1 - \rho_{x0})(1 - \rho_{t0})}$$

Also,

$$P(X^* = 1) \; = \sum_z P(X^* = 1 | Z = z) P(Z = z)$$

$$= (\rho_{x1} + (1 - \rho_{x0})) \cdot 0.5.$$

$$P(X^* = 0) = 1 - P(X^* = 1) = (\rho_{x0} - \rho_{x1} + 1) \cdot 0.5.$$

With the above expressions, we can check that setting $\rho_{x0} = 0.1$, $\rho_{t0} = 0.2$, $\rho_{x1} = 0.3$, $\rho_{t1} = 0.4$ gives a non-zero ATE when computed using non-causal methods (i.e. without accounting for hidden confounder). This corresponds to the non-causal baseline in Table 1. To get ATE of zero using non-causal baseline, we need to set $\rho_{t1} = 1 - \rho_{t0}$ and $\rho_{x1} = 1 - \rho_{x0}$.

# B  Neural Architecture of Deep Structural Equations and Approximate Inference Networks

**Our Architecture**   In this section, we add the details of the DGSE and DMSE architecture that we used. $\mathbf{X}_i$ denotes an input datapoint, i.e. the feature vector (possibly containing multiple modalities), $t_i$ is the treatment assignment, $y_i$ denotes the corresponding outcome and $\mathbf{Z}_i$ is the latent hidden confounder. Within DGSE and DMSE, the latent variable is modeled as a Gaussian. For DGSE, we write (similar to Louizos et al. [28]):

$$p[\mathbf{Z}_i] = \prod_{j=1}^{D_z} \mathcal{N}(Z_{ij} \mid 0, 1)$$

$$p[t_i \mid \mathbf{Z}_i] = \mathsf{Bern}(\sigma(\mathrm{NN}_1(\mathbf{Z}_i)))$$

$$p[\mathbf{X}_i \mid \mathbf{Z}_i] = \prod_{j=1}^{D_x} p[X_{ij} \mid \mathbf{Z}_i]$$

where $\sigma(\cdot)$ is the sigmoid function, Bern is the Bernoulli distribution, $D_x, D_z$ are the dimensions of $\mathbf{X}$ and $\mathbf{Z}$ respectively, and $p[X_{ij} \mid \mathbf{Z}_i]$ is an appropriate probability distribution for the covariate $j$. If the treatment variable is not binary, we can modify the distribution appropriately. Within DMSE, it is possible to further factorize the distribution $p(X_i|Z_i)$ into product of distributions over component modalities owing to the conditional independence.

If the outcome $y$ is discrete, we parameterize its probability distribution as a Bernoulli distribution:

$$p[y_i \mid t_i, \mathbf{Z}_i] = \mathsf{Bern}\,(\pi = \hat{\pi}_i)$$

$$\hat{\pi}_i = \sigma\,(\mathrm{NN}_2(\mathbf{Z}_i, t_i))$$

and if it is continuous, we parameterize its distribution as a Gaussian with a fixed variance $\hat{v}$, defined as:

$$p[y_i \mid t_i, \mathbf{Z}_i] = \mathcal{N}\,(\mu = \hat{\mu}_i, \sigma^2 = \hat{v})$$

$$\hat{\mu}_i = \mathrm{NN}_2(\mathbf{Z}_i, t_i).$$

Here each of the $\mathrm{NN}_i(\cdot)$ is a neural network.

The posterior distribution for DGSE is approximated as

$$q[\mathbf{Z}_i \mid \mathbf{X_i}, t_i, y_i] = \prod_{j=1}^{D_z} q[Z_{ij} \mid \mathbf{X_i}, t_i, y_i] = \prod_{j=1}^{D_z} \mathcal{N}(\mu_{ij}, \sigma_{ij}^2),$$

where

$$\mu_{ij}, \sigma_{ij}^2 = \text{NN}_4(\mathbf{X}_i, y_i, t_i).$$

For DMSE, the posterior distribution is computed differently using Product-of-Experts (PoE) [56] formulation, due to which it can handle missing modalities during training and inference gracefully.

The objective of DGSE model is the variational lower bound defined as:

$$\mathcal{L} = \sum_{i=1}^{N} \mathbb{E}_{q[\mathbf{Z}_i \mid \mathbf{X}_i, t_i, y_i]} \Big[ \log p[\mathbf{Z}_i] + \log p[\mathbf{X}_i, t_i \mid \mathbf{Z}_i]$$
$$+ \log p[y_i \mid \mathbf{Z}_i, t_i] - \log q[\mathbf{Z}_i \mid \mathbf{X}_i, t_i, y_i] \Big]$$

The DMSE, on the other hand, requires a sub-sampled training objective to ensure that the modality specific posterior networks are trained and the relationships between individual modalities is captured. For DGSE, we also define the auxiliary encoders and the extra term in the variational lower bound following Louizos et al. [28].

Auxiliary Encoders:

$$q[t_i \mid \mathbf{X}_i] = \text{Bern}(\pi = \sigma(\text{NN}_5(t_i)))$$

For discrete $y_i$, we have

$$q[y_i \mid t_i, \mathbf{X}_i] = \text{Bern}(\pi = \hat{\pi}_i)$$
$$\hat{\pi}_i = \sigma(\text{NN}_6(\mathbf{X}_i, t_i)).$$

For continuous $y_i$, we write

$$p[y_i \mid t_i, \mathbf{X}_i] = \mathcal{N}\left(\mu = \bar{\mu}_i, \sigma^2 = \hat{v}\right)$$
$$\hat{\mu}_i = \text{NN}_6(\mathbf{X}_i, t_i).$$

This introduces the following extra term in the variational lower bound:

$$\mathcal{L}' = \sum_{i=1}^{N} \log q[t_i \mid \mathbf{X}_i] + \log q[y_i \mid \mathbf{X}_i, t_i]\Big]$$

DMSE does not involve these extra terms within its ELBO objective.

Compared with Louizos et al. [28], we can extend DGSE to different types of architectures for the posterior distribution $q[\mathbf{Z}_i \mid \mathbf{X_i}, t_i, y_i]$. When $\mathbf{X}$ is an image (e.g. medical scans, patient photos), we can use a suitable Convolutional Neural Network (CNN) architecture for extracting information effectively [23]. In our experiments with image modality, we used pretrained CLIP embeddings [24] in the first layer to extract relevant features from the images. To avoid the overwhelming difference between the image and two binary variables $t, y$, we also apply dimension reduction techniques such as Principle Component Analysis to the embeddings of the image before feeding it into the network that is shared with $t, y$. When $\mathbf{X}$ is time-series data, (e.g. text, recording), we can change the architecture to recurrent neural networks such as Long Short Term Memory [14]. More generally, we can choose modality specific architectures and make appropriate design choices to perform learning and inference over unstructured modalities as inputs. DMSE can handle different types and lengths of modalities gracefully and also work with missing modalities owing the specialized variational learning and inference procedures.

## C   Comparing Our Methods with Other VAE- Based Estimators

While our method is an instance of generative models, we identify the following key differences:

1. We propose **new generative model architectures** that extend existing models (e.g., DSE, CEVAE) to multiple proxies $X_i$, each possibly coming from a different modality.

2. We derive **novel inference algorithms** for these extended models, which have the following benefits:

(a) Our algorithms scale better to large sets of modalities by leveraging the independence structure of the $X_i$.

(b) Our inference algorithms naturally handle missing $X_i$.

(c) They are also simpler: they don't require auxiliary networks (e.g., like in CEVAE [28]).

3. Lastly, our key contribution is that we demonstrate the effectiveness of generative models at **modeling unstructured proxies** (many previous methods instead relied on propensity scoring).

Appendix H.5.1 empirically shows that DMSE model compares favorably against CEVAE on synthetic datasets.

## D Setups used for IHDP and STAR Dataset experiments

### D.1 IHDP Experiments

The data corresponding to non-white mothers in the treated set of children is removed so that causal effect of the intervention cannot be estimated directly. The column corresponding to mother's race is removed so that this confounder cannot be obtained directly from the input. We consider 100 replicates of this dataset, where the output is simulated according to setting 'A' of NPCI package [9]. The true treatment effect is known as the simulation provides expected output values for both values of binary treatment variable. We train a DGSE model on each replicate with a 63/27/10 ratio of training, validation and test dataset size. We set the latent dimension to be 20 units and the number of hidden layers to be 2. The hidden layers have size of 20 units.

**The IHDP-Full Setting**   There are 25 input features in this experimental setting. We report the absolute error in ATE produced by DGSE and OLS for this setting in Table 7.

**The IHDP-Mini Setting**   Here, we choose 9 features from the 25 input features so that removal of the feature 'baby's gender' produces statistically significant treatment effect. We used mutual information and F-statistics between each of the original 25 features and the target variable $y$ to assess the importance of each feature in the initial 100 replicates of IHDP. While making sure that the absolute ATE errors don't deviate too much from the corresponding errors produced by IHDP-Full setting, we experimented with several combinations of the high ranking features to select the following 9 features in the IHDP-Mini setting.

1. Feature 6: 'sex of baby'
2. Feature 0: 'birth-weight'
3. Feature 1: 'b.head'
4. Feature 2: 'preterm'
5. Feature 3: 'birth.o'
6. Feature 8: 'mom married?'
7. Feature 9: 'mom's education lower than high school?'
8. Feature 12: 'Smoked cig during pregnancy?'
9. Feature 20: 'harlem'

Table 7 shows a comparison of absolute ATE errors between the IHDP-Full setting and IHDP-Mini setting. Table 2 shows the comparison of our approach with CFRNet [16] and Ordinary Least Squares (OLS) approach. OLS takes the concatenation of the covariates and treatment variable value as input to produce output. We see that the CFRNet baseline does not utilize the image effectively while OLS shows small difference between the setting where baby's gender was removed (8 attrs) and the setting where baby's gender was retained (9 attrs) setting. Hence the replacement of image (8 attrs+image) produces a small average improvement as compared to the setting where baby's gender was dropped (8 attrs). We also note that we have dropped 6 replicates from the 100 IHDP replicates under consideration. These 6 replicates showed a large degradation in ATE estimates by adding baby's gender (9 attrs) as compared to removing it (8 attrs).

### D.2 STAR Experiments

We 'derandomize' this dataset by removing 80% of the data corresponding to white students in the treated population. The dataset [1] has 15 input attributes. The true treatment effect can be estimated directly since the original dataset corresponds to a randomized controlled trial. Hence, it is possible to compute ATE error as the absolute difference between true ATE estimate and the ATE as predicted by the model. Similar to IHDP, we choose the following subset of attributes for the STAR experiment

1. Feature 2: 'Student grade'
2. Feature 3: 'Student class-type'
3. Feature 4: 'Highest degree obtained by teacher'
4. Feature 5: 'Career ladder position of teacher'
5. Feature 6: 'Number of years of experience of teacher'
6. Feature 7: 'Teacher's race'
7. Feature 10: 'Student's gender'
8. Feature 11: 'Student's ethnicity'

Table 6: Multimodal Experiments on STAR Dataset: Removing student gender and ethnicity (6 attrs) shows increased ATE errors when compared with retaining these attributes (8 attrs), signaling that these two attributes are important for predicting treatment effect. Replacing these with image of a child shows no degradation in ATE estimation.

| Setting | $\varepsilon_{ATE}$ (Train+Val) | $\varepsilon_{ATE}$ error(Test) |
|---|---|---|
| Deep Str Eqns | | |
| 8 attrs | 36.479 (1.770) | 34.039 (2.336) |
| 6 attrs | 43.682 (1.520) | 40.651 (2.425) |
| 6 attrs + image | 35.581 (1.723) | 33.654 (2.476) |
| CFRNet | | |
| 8 attrs | 61.835 (1.025) | 25.436 (2.332) |
| 6 attrs | 62.055 (1.001) | 25.649 (2.339) |
| 6 attrs + image | 61.350 (1.109) | 25.219 (2.313) |

In Table 6, we repeat the experiment 100 times and report average ATE errors along with standard error. We removed 8 repetitions in the experiment where DGSE or CFRNet showed lack of convergence as evidenced by very high validation loss on any setting of input attributes.

Table 7: Treatment effects on IHDP Dataset. Using a reduced set of features in the IHDP Mini setting produces comparable absolute ATE errors as the degradation is small. Numbers in round braces indicate standard deviations. Since this is a simulated dataset, we can directly compute the treatment effect using the simulated factual and counterfactual outputs. ATE error is the absolute difference between true ATE and predicted ATE.

| MODELS | INPUT | $\varepsilon_{ATE}$ (TRAIN+VAL) | $\varepsilon_{ATE}$ ERROR(TEST) |
|---|---|---|---|
| DGSE | FULL | 0.289 (0.027) | 0.358 (0.041) |
| OLS | FULL | 0.535 (0.089) | 0.718 (0.132) |
| DGSE | MINI | 0.404 (0.107) | 0.720 (0.159) |

## E  Evaluating Quality of Pre-Trained Embeddings

We demonstrate that our pre-trained embeddings contain useful signal by building a neural network model that predicts the gender, age and ethnicity from the CLIP embedding [24] of the corresponding image.

We build a simple neural network model that takes the CLIP embedding of an image as input and predicts the age of the person in that image. We use 5-dimensional PCA embeddings of 500 randomly chosen images of people aged 10-45 years (corresponding to the age-group in the IHDP experiment). We have an independent test dataset corresponding to 100 images chosen in a similar way. We see that the $R^2$ value for the age prediction on test dataset is 0.45. If we increase the size of PCA embedding to 50, this $R^2$ value increases to 0.58. Thus, it is possible to extract the age information from randomly chosen images using a simple neural network.

In the above setting, we also studied the classification accuracy of separate neural networks that predict gender and ethnicity from the CLIP embeddings. We saw that gender was predicted with 94% accuracy and ethnicity was predicted with 58% accuracy using a 5-dimensional PCA embedding. After increasing the size of PCA embedding to 50 dimensions, the gender prediction accuracy increased to 95% and ethnicity prediction accuracy increased to 77%. This further supports our idea of replacing the attribute corresponding to 's baby's gender or student ethnicity/gender with an appropriate image.

# F   Plant GWAS

**Setup**   We apply our deep structural equations framework for correcting the effects of confounding. We fit a DGSE model in two stages: (1) first, we only fit the model of $p(T|Z)$ using the DGSE ELBO objective; (2) then we fit $p(Y|Z,T)$ with a fixed $Z$ produced by the auxiliary model $q(Z|T)$. We found this two-step procedure to produce best results. The subsampled SNPs corresponding to each genome are taken as input $X$. The encoder and decoder use a single hidden layer of 256 units while a 10-dimensional latent variable $Z$ is used. This network is optimized using ClippedAdam with learning rate of 0.01, further reduced exponentially over 20 training epochs. The confounding variable for each genome can now be computed as latent representation produced by the DSE. To measure the success of confounding correction, we compute the $R^2$ values between the true GDP of the region and the GDP output as predicted via our model and the baselines. If we have corrected for confounding, then we should get low $R^2$ values.

**Historical Weather Data**   We used historical weather data collected from Menne et al. [32] to add a new modality while performing plant GWAS. We use per day precipitation data from year 2000 collected by weather station closest in distance to the latitude/longitude coordinates of the location from which the SNPs of plant were collected. For the locations where weather data was missing, we replaced those entries with zeros.

# G   Simulated GWAS Experiments

We provide additional details on this experiment here.

## G.1   Data generating process

To simulate the confounders, SNPs (genotypes), and the outcomes (phenotypes), we follow the "Spatial" simulation setup from Appendix D.1 & D.2 of [49]. Specifically, we generate random low-rank factorization of the allele frequency logits $F = \sigma^{-1}(\Gamma S)$, where $\sigma$ is sigmoid, as is common in the literature [4, 40]. In the "Spatial" Setting [49], the $S$ matrix is interpreted as geographic spatial positions of the individuals. For the $m$-th SNP of the $n$-th individual, we generate the SNP $X_{nm} \sim \text{Bin}(3, \sigma(F_{nm}))$. In this simulation, we considered $M = 10$ SNPs with $N = 10000$ individuals.

Individuals are clustered into $K = 3$ groups based on their locations, and the individual's cluster is the *unobserved* confounder. Then, the outcomes are calculated from *both* the SNPs, where only $c = 2$ SNPs have a non-zero causal effect, plus a confounding term that is a function of the cluster, and some i.i.d. Gaussian noise. One small deviation from Tran & Blei [49] is that our noise is i.i.d. Gaussian (hence, the variance does not depend on the confounder).

We further augmented the dataset to include a time-series proxy that can help identify the unobserved spatial position of individuals. For each cluster, we came up with some Fourier coefficients,

$$\text{Cluster 0: } a_0^{(0)} = 0.0, a_1^{(0)} = -1.0, a_2^{(0)} = 1.0, b_1^{(0)} = -1.0, b_2^{(0)} = 1.0$$
$$\text{Cluster 1: } a_0^{(1)} = 1.0, a_1^{(1)} = -5.0, a_2^{(1)} = 2.0, b_1^{(1)} = -5.0, b_2^{(1)} = 2.0$$
$$\text{Cluster 2: } a_0^{(2)} = -1.0, a_1^{(2)} = -2.0, a_2^{(2)} = 5.0, b_1^{(2)} = -2.0, b_2^{(2)} = 5.0.$$

For each individual in the $k$-th cluster, we sampled at $N_{samples} = 50$ uniformly spaced times, across $N_{periods} = 2$ periods of length $T = 5$. That is, our time series proxy consists of points $\{x_i\}_{i \in [N_{samples}]}$,

$$x_i = \frac{1}{2}a_0^{(k)} + \sum_{\ell=1}^{2} a_\ell^{(k)} \cos\left(\frac{2\pi t_i(\ell-1)}{T}\right) + \sum_{\ell=1}^{2} b_\ell^{(k)} \sin\left(\frac{2\pi t_i(\ell-1)}{T}\right),$$
$$t_i = \frac{T N_{periods} i}{N_{samples}}.$$

Please see Fig. 3 for a visualization of the time-series generated for each cluster.

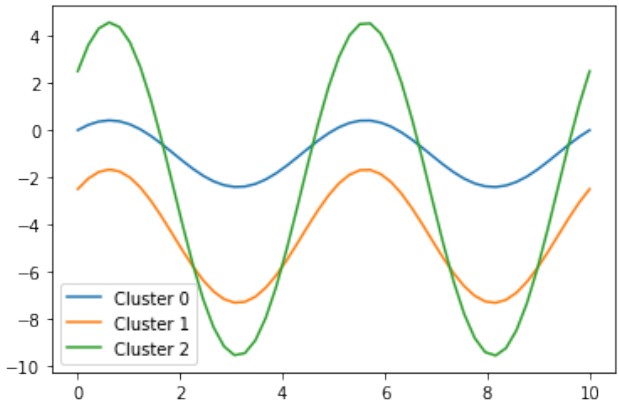

Figure 3: Visualization of Time-series Proxy.

## G.2 DSE Modeling and evaluation setup

To handle the SNP data, we trained a Multi-Layer Perceptron (MLP) based $\beta$-variational autoencoder [12] with ReLU activations, which output parameters of a Gaussian latent distribution. The MLPs consisted of 3 hidden layers, each with 64 units, and our latent dimension $z$ was 2. To handle time series data (new modality), we trained a $\beta$-VAE that encoded and decoded with 1D convolutions, as 1D convolutions have been shown to strongly approximate fast Fourier transforms [34]. There were 3 hidden convolutional layers, each of which had 32 output channels, with a kernel of length 3 and a stride of 1. Following the convolution was an MLP with 1 hidden layer consisting of 64 units. ReLU activations were used throughout the architecture.

In our experiments, we swept over $\beta \in \{0.1, 0.2, 0.5, 1.0, 1.2, 1.5\}$. Our general hypothesis was that lower $\beta$ values would make our VAE perform better, as unlike in the standard VAE evaluation scheme, we sample from the posterior to generate latents for individuals, and not the prior, so the KL divergence term in the ELBO doesn't matter as much.

When evaluating on a set of SNPs and time series data corresponding to a set of individuals, we generate latents by passing the data of each respective type into the appropriate encoder and sampling from the resulting distributions. The latent $z$ for each individual is the concatenation of the SNP latent $z_{\text{snp}}$ and the time series latent $z_{\text{ts}}$.

Once we have the latent confounder $z$ for an individual, we calculate the causal effect of the $m^{\text{th}}$ SNP by performing the linear regression,

$$y = \beta_0 + \sum_i \beta_i z_i + \gamma_m x_m + \epsilon_i,$$

where $\epsilon_i$ are i.i.d. standard normal. We solved this using the closed form solution for least-squares linear regression, given by

$$\widehat{\gamma} = (A^T A)^{-1} A^T y,$$

where $A_i = [z_i, 1, x_m]$ is the $i^{\text{th}}$ row in the matrix $A$. Then $\widehat{\gamma}_m$ is our estimated causal effect for the $m^{\text{th}}$ SNP. We let $\widehat{\gamma}$ be a vector of length $M$ to denote our estimates for each SNP's causal effect.

### G.3 Results

We tested our single modality DSE model and multi-modal DSE model against several linear latent model baselines. Two baselines, Principle Components Analysis (PCA) and Factor Analysis (FA), also generate a latent $z$, which we then use within our linear regression approach to calculate $\widehat{\gamma}$. We also ran the Linear Mixed Model (LMM) implementation by Limix [27]. We also plot two oracle baselines. The first is when $\widehat{\gamma} = \gamma^\star$, labeled "truth". The second is when $z$ is the true confounder used when generating the data, according to "Spatial".

Let $\gamma^\star$ denote the ground truth causal effect vector. We report $\ell_1$ and $\ell_2$ norms to measure $\|\widehat{\gamma} - \gamma^\star\|$. We also report true/false positive/negatives, which we define as follows. For an individual, let $\tau = \min_{i:\gamma_i \neq 0} |\gamma_i|/2$. Then, SNP $m$ is a

1. True Positive (tp): if $\widehat{\gamma}_m \geq \tau$ and $\gamma_m^\star \geq \tau$ or $\widehat{\gamma}_m \leq -\tau$ and $\gamma_m^\star \leq -\tau$.
2. True Negative (tn): if $|\widehat{\gamma}_m| < \tau$ and $|\gamma_m^\star| < \tau$.
3. False Negative (fn): if $|\widehat{\gamma}_m| < \tau$ but $|\gamma_m^\star| > \tau$.
4. False Positive (fp): if none of the above hold. Concretely, there are two cases. First, if $|\widehat{\gamma}_m| > \tau$ but $|\gamma_m^\star| < \tau$. Second, if the direction is wrong, i.e. $\widehat{\gamma}_m > \tau$ but $\gamma_m^\star < -\tau$, or $\widehat{\gamma}_m < -\tau$ but $\gamma_m^\star > \tau$.

Finally, recall that precision is $tp/(tp + fp)$ and recall is $tp/(tp + fn)$, and higher is better for both. In the following table, we show the mean and standard error of the mean (sem) over 10 seeds at evaluation time. The unimodal DSE only encodes the SNPs vector into a 2 dimensional embedding. The multimodal DSE also encodes the time-series vector into a 2 dimensional embedding, and we concatenate this to the 2-dimensional embedding of SNPs to form a 4 dimensional embedding that can be used in our linear regression. $PCA_k$ denotes Principal Components Analysisn(PCA) with $k$ components. Similarly, $FA_k$ denotes Factor Analysis (FA) with $k$ components. Amongst the non-oracle baselines, multi-modal DSE has the smallest errors and highest precision/recall. It beats all linear baselines including LMM. Unimodal DSE, while worse than LMM, still beats PCA and FA. In general, we see that precision starts to deteriorate faster than recall, suggesting that false positives are more likely from the weaker linear deconfounding techniques such as PCA/FA. It's also interesting that when PCA/FA used more components, we see worse performance across the board. This illustrates the importance of picking the right size of latent dimension, which is an open question. If the latent dimension is too small then we may not capture all the confounders. If the latent dimension is too large, the latent encoding may also end up capturing spurious correlations.

## H  Alternative Causal Graph Structures

This section derives estimators of ATE and ITE under assumptions on the causal graph that are different from the ones described in Figure 1. The key takeaway of this section is that ATE and ITE can be computed under arbitrary causal graphs using minor modifications of our algorithms, which we describe in this section.

### H.1  Alternative Causal Graph Structures

We start by formally defining the class of causal graphs that will be studied in this section. Formally, we identify two sets of modification to Figure 1. We will derive small modifications to our algorithms to allow the new types of causal graphs implied by these modifications.

| Model | $\ell_1(\downarrow)$ Mean (sem) | $\ell_2(\downarrow)$ Mean (sem) | tp ($\uparrow$) Mean (sem) | fp ($\downarrow$) Mean (sem) | tn ($\uparrow$) Mean (sem) | fn ($\downarrow$) Mean (sem) |
|---|---|---|---|---|---|---|
| Optimal | 0.22 (0.04) | 0.09 (0.02) | 2.0 (0.00) | 0.1 (0.10) | 7.9 (0.10) | 0.0 (0.00) |
| DSE (2 modalities) | 0.30 (0.06) | 0.14 (0.03) | 2.0 (0.00) | 0.2 (0.13) | 7.8 (0.13) | 0.0 (0.00) |
| LMM | 0.44 (0.06) | 0.17 (0.02) | 2.0 (0.00) | 0.7 (0.42) | 7.3 (0.42) | 0.0 (0.00) |
| DSE (1 modality) | 0.60 (0.09) | 0.26 (0.05) | 2.0 (0.00) | 1.1 (0.59) | 6.9 (0.59) | 0.0 (0.00) |
| PCA (1 component) | 0.93 (0.17) | 0.40 (0.07) | 2.0 (0.00) | 2.3 (0.72) | 5.7 (0.72) | 0.0 (0.00) |
| FA (1 component) | 1.08 (0.17) | 0.59 (0.13) | 2.0 (0.00) | 2.0 (0.71) | 6.0 (0.71) | 0.0 (0.00) |
| PCA (2 components) | 1.38 (0.24) | 0.56 (0.09) | 1.8 (0.13) | 3.4 (0.83) | 4.6 (0.83) | 0.2 (0.13) |
| FA (2 components) | 1.44 (0.30) | 0.71 (0.19) | 2.0 (0.00) | 2.6 (0.75) | 5.4 (0.75) | 0.0 (0.00) |
| PCA (3 components) | 1.66 (0.23) | 0.68 (0.09) | 1.6 (0.16) | 3.8 (0.80) | 4.2 (0.80) | 0.4 (0.16) |
| FA (3 components) | 1.89 (0.45) | 0.95 (0.25) | 1.8 (0.13) | 3.0 (0.56) | 5.0 (0.56) | 0.2 (0.13) |

Table 8: Comparison of DSE with Baselines to Perform GWAS.

| Model | Precision ($\uparrow$) Mean (sem) | Recall ($\uparrow$) Mean (sem) |
|---|---|---|
| Optimal | 0.97 (0.03) | 1.0 (0.00) |
| DSE (2 modalities) | 0.93 (0.04) | 1.0 (0.00) |
| LMM | 0.85 (0.08) | 1.0 (0.00) |
| DSE (1 modality) | 0.78 (0.08) | 1.0 (0.00) |
| PCA (1 component) | 0.58 (0.09) | 1.0 (0.00) |
| FA (1 component) | 0.62 (0.08) | 1.0 (0.00) |
| PCA (2 components) | 0.44 (0.09) | 0.9 (0.07) |
| FA (2 components) | 0.55 (0.09) | 1.0 (0.00) |
| PCA (3 components) | 0.37 (0.08) | 0.8 (0.08) |
| FA (3 components) | 0.44 (0.08) | 0.9 (0.07) |

Table 9: Comparison of DSE with Baselines in Terms of Precision and Recall Metrics.

**Observed confounders**   We consider an expanded set of graphs over a space of random variables $(X, Y, T, Z, V)$, where $V$ represents observed (non-proxy) confounders and the other random variables are associated with the observed data $x, y, t, z$. We look at causal graphs implied by structural equations of the form:

$$Z \sim \mathcal{P}_Z \quad V \sim \mathcal{P}_V \quad X_j \sim \mathcal{P}_{X_j}(\theta_{X_j}(Z)) \;\; \forall j \quad T \sim \mathrm{Ber}(\pi_T(Z, V)) \quad Y \sim \mathcal{P}_Y(\theta_Y(Z, V, T))), \tag{8}$$

where $\mathcal{P}_{X_j}, \mathcal{P}_Y$ are probability distributions with a tractable density and the $\mu, \sigma, \pi, \theta$ are functions parameterized by neural networks that output the parameters of their respective probability distribution as a function of ancestor variables in the causal graph.

In the above equations, $V$ is variable that is assumed to be **always observed** (just like $y^{(i)}$ and $t^{(i)}$). All the other technical terms are defined as in the main body of the paper. The above equations result in a causal graph with edges between $V$ and $Y, T$ and define a distribution $p(x, y, t, z, v)$.

**Dependent proxies**   Another possible set of modifications to Figure 1 is the presence of edges between proxies $X_i, X_j$, which can be denoted as

$$X_j \sim \mathcal{P}_{X_j}(\theta_{X_j}(Z, \mathrm{pa}(X_j)))) \;\; \forall j$$

where $\mathrm{pa}(X_j))$ denotes the set of parents of $X_j$ among the other unstructured proxy variables $X_i$.

## H.2   A General Estimator Class

The following Theorem shows that we can estimate ATE and ITE when the data distribution follows the structure in Figure 1, plus the two types of modifications outlined above (observed confounders and dependent proxies).

**Theorem 3.** *The true ITE$(x, \mathcal{M})$ for any subset $\mathcal{M} \subseteq \{1, 2, ..., m\}$ of observed modalities is identifiable when the true data distribution $p(x, y, t, z, v)$ has the causal graph structure of a DMSE model (Figure 1) in addition to having observed confounders $v$ and possibly dependent confounders.*

**Proof:** Let $x_{\mathcal{M}} = \{x_j \mid j \in \mathcal{M}\}$ be the data from the observed subset of modalities. Let $v$ be the observed proxy variable. We need to show that $p(y|x_{\mathcal{M}}, v, \text{do}(t = t'))$ is identifiable for any $t'$. Observe that

$$p(y|x_{\mathcal{M}}, v, \text{do}(t = t')) = \int_z p(y|z, x_{\mathcal{M}}, v, do(t = t'))p(z|x_{\mathcal{M}}, v, do(t = t'))dz$$

$$= \int_z p(y|z, x_{\mathcal{M}}, v, t')p(z|x_{\mathcal{M}})dz,$$

where the second equality follows from the rule of do-calculus (applying backdoor adjustment). Since our proof holds for any $t'$ and all elements on the right-hand side are identifiable, the claim follows. ∎

### H.3 Observed Confounders

Next, we derive an extension of the DMSE model to the setting in which we have observed confounders $V$. We refer to this modified model as **DMSE-V**.

As earlier, the **DMSE-V** model induces a tractable joint density $p(X, Y, T, Z, V)$, which allows us to fit its parameters using stochastic variational inference by optimizing the evidence lower bound (ELBO) on the marginal log-likelihood $p(y^{(i)}, x^{(i)}, t^{(i)}, v^{(i)})$ defined over an expanded dataset $\{y^{(i)}, x^{(i)}, t^{(i)}, v^{(i)}\}_{i=1}^n$:

$$\text{ELBO}_X = \sum_{i=1}^n \mathbb{E}_q \left[ \sum_{j=1}^m \log p(x_j^{(i)}|z) + \log p(y^{(i)}, t^{(i)}, v^{(i)}, z) - \log q(z|x^{(i)}, y^{(i)}, t^{(i)}, v^{(i)}) \right], \tag{9}$$

where $p(y^{(i)}, t^{(i)}, v^{(i)}, z) = p(y^{(i)}|t^{(i)}, v^{(i)}, z)p(t^{(i)}|z, v^{(i)})p(z)p(v^{(i)})$ and $q(z|x^{(i)}, y^{(i)}, t^{(i)}, v^{(i)})$ is the approximate variational posterior. Note that since $v^{(i)}$ *is always observed* (just like $y^{(i)}$ and $t^{(i)}$), the $p(v^{(i)})$ term can be ignored.

In practice, this reduces to the vanilla DMSE model with the following modifications:

- The $\log p(y^{(i)}|t^{(i)}, v^{(i)}, z)$ term becomes additionally conditioned on $v^{(i)}$.
- The $\log p(t^{(i)}|v^{(i)}, z)$ term becomes additionally conditioned on $v^{(i)}$.
- The approximate posterior $q(z|x^{(i)}, y^{(i)}, t^{(i)}, v^{(i)})$ becomes additionally conditioned on $v^{(i)}$.

Crucially, the specialized inference algorithms derived for the DMSE model remain unchanged. Since $V$ is always observed, learning a **DMSE-V** model is equivalent to learning a model $p(y, x, t, z|v)$, which has the same structure as a DMSE model. In particular, all the $X_i$ are conditionally independent given $Z$. Hence, the same learning and inference algorithms apply.

### H.4 Causal Links Among Proxies

Another type of causal graph that we consider is one in which proxies $X_i$ are connect by causal edges. First, we note that when proxies are unstructured, such causal edges are expected to be rare, i.e., we do not expect the pixels of an image $X_i$ to have a direct influence on other variables.

When $X_i$ takes on a structured form and directly influences other proxies, our strategy is to "collapse" any sets of variables $X_j$ that have edges among them, until we have the conditionally independent structure in Figure 1. In the extreme case, we might need to collapse all proxies $X_i$ into a single proxy $X$ that consists of their concatenation.

The result is a model that has the same structure as DMSE, and that can be learned using the same set of inference and learning algorithms. The only drawback is increased computational efficiency.

## H.5 Additional Synthetic Data Experiment Details

We generate synthetic data to simulate a process with modified causal links as covered in subsection H.3 and H.4. Specifically, we simulate four different datasets with generative process detailed in the following section. We generate $\{X_i, t_i, y_i\}_{i=0}^{i=m}$ where $m = 10000$. We use train/val/test split of 63/27/10. Here, $\oplus$ stands for logical XOR.

1. **Case of Original Causal Graph (Dataset A)**

   Variables: $X1$ is unstructured input, $X2$ is structured input, $T$ is treatment, $Y$ is output, $Z$ is confounder

   Edges in graph: $Z \to X1, Z \to X2, \{Z\} \to T, \{T, Z\} \to Y$

   Thus we added no extra edges.

   Generative process:

   (a) $P(z = 1) = 0.5$

   (b) $P(x1' = 1|z = 1) = P(x1' = 0|z = 0) = 0.1$ ($x1'$ is an intermediate variable)

   (c) $P(x2 = 1|z = 1) = P(x2 = 0|z = 0) = 0.2$

   (d) $P(t = 1|z = 1) = P(t = 0|z = 0) = 0.2$

   (e) $y = (z \oplus t)$

   (f) $P(x1|x1' = 1)$ is unif. over MNIST images of '1', $P(x1|x1' = 0)$ is unif. over MNIST images of '0'

2. **Case of the Observed Confounder (Dataset B)**

   Variables: $X1$ is unstructured input, $X2$ is structured input, $T$ is treatment, $Y$ is output, $Z$ is confounder

   Edges in graph: $Z \to X1, \{X2, Z\} \to T, \{X2, T, Z\} \to Y$

   Thus we added extra edges $X2 \to T$ and $X2 \to Y$.

   Generative process:

   (a) $P(z = 1) = 0.5, P(x2 = 1) = 0.5$

   (b) $P(x1' = 1|z = 1) = P(x1' = 0|z = 0) = 0.1$ ($x1'$ is an intermediate variable)

   (c) $P(t = 1|z = 1, x2 = 1) = P(t = 0|z = 0, x2 = 1) = 0.2$;
   $P(t = 1|z = 1, x2 = 0) = P(t = 0|z = 0, x2 = 0) = 0.9$

   (d) $y = x2$ AND $(z \oplus t)$

   (e) $P(x1|x1' = 1)$ is unif. over MNIST images of '1', $P(x1|x1' = 0)$ is unif. over MNIST images of '0'

3. **Case of the Observed Confounder (Dataset C)**

   Variables: $X1$ is unstructured input, $X2$ is structured input, $T$ is treatment, $Y$ is output, confounder $Z = (Z1, Z2)$

   Edges in graph: $Z \to X1, Z \to X1, Z-> X2, \{X2, Z\} \to T, \{X2, T, Z\} \to Y$

   Thus we added extra edges $Z \to X2, X2 \to T$ and $X2 \to Y$.

   Generative process:

   (a) $P(z1 = 1) = 0.5, P(z2 = 1) = 0.5$

   (b) $P(x1' = 1|z1 = 1) = P(x1' = 0|z1 = 0) = 0.1$ ($x1'$ is an intermediate variable)

   (c) $P(x2 = 1|z2 = 1) = P(x2 = 0|z2 = 0) = 0.9$

   (d) $z = z_1 \oplus z_2$

   (e) $P(t = 1|z = 1, x2 = 1) = P(t = 0|z = 0, x2 = 1) = 0.9; P(t = 1|z = 1, x2 = 0) = P(t = 0|z = 0, x2 = 0) = 0.1$

   (f) $y = x2 \oplus t \oplus (z1 \oplus z2)$

   (g) $P(x1|x1' = 1)$ is unif. over MNIST images of '1', $P(x1|x1' = 0)$ is unif. over MNIST images of '0'

4. **Case of Causal Links Among Proxies (Dataset D)**

   Variables: $X1$ is structured input, $X2$ is structured input, $X3$ is unstructured input, $T$ is treatment, $Y$ is output, confounder $Z = (Z1, Z2, Z3)$

   Edges in graph: $Z \to X1, Z \to X2, Z \to X3, X1 \to X2, \{Z\} \to T, \{T, Z\} \to Y$

   Thus we added extra edge: $X1 \to X2$

   Data generation:

(a) $P(z1 = 1) = P(z2 = 1) = P(z3 = 1) = 0.5$

(b) $P(x1 = 1|z1 = 1) = P(x1 = 0|z1 = 0) = 0.1$

(c) $P(x2 = 1|z2 = 1, x1 = 0) = P(x2 = 0|z2 = 0, x1 = 0) = 0.8$
$P(x2 = 1|z2 = 1, x1 = 1) = P(x2 = 0|z2 = 0, x1 = 1) = 0.2$

(d) $P(x3' = 1|z3 = 1) = P(x3' = 0|z3 = 0) = 0.3$ ($x3'$ is an intermediate variable)

(e) $z = z_1 \oplus z_2 \oplus z_3$

(f) $P(t = 1|z = 1) = P(t = 0|z = 0) = 0.2$

(g) $y = t \oplus z$

(h) $P(x3|x3' = 1)$ is unif. over MNIST images of '1', $P(x3|x3' = 0)$ is unif. over MNIST images of '0'

5. **Case of Increasing Number of Proxies (Dataset E)**

   Variables: $\{X_1, X_2, ..X_m\}$ are unstructured inputs, $T$ is treatment, $Y$ is output, confounder $Z = (Z_1, Z_2, .., Z_m)$. $m$ is number of modalities Edges in graph: $Z_i \rightarrow X_i, Z \rightarrow T, \{T, Z\} \rightarrow Y$

   (a) $P(z_i = 1) = \frac{i}{m}$

   (b) $P(x_i' = 1|z_i = 1) = P(x_i' = 0|z_i = 0) = 1$ ($x_i'$ are intermediate variables)

   (c) $z = \oplus_{i=1}^{m} z_i$ (xor over all $z_i$)

   (d) $P(t = 1|z = 1) = P(t = 0|z = 0) = 0.25$

   (e) $P(y = 1|t = 1) = \text{sigmoid}(3z + 2), P(y = 1|t = 0) = \text{sigmoid}(3z - 2)$

   (f) $P(x_i|x_i' = 1)$ is unif. over MNIST images of '1', $P(x_i|x_i' = 0)$ is unif. over MNIST images of '0'

Table 10: Comparison of DMSE and CEVAE with increasing number of proxies

| NUMBER OF INPUT MODALITIES | CEVAE | | DMSE | | % IMPROVEMENT MADE BY DMSE W.R.T CEVAE | |
|---|---|---|---|---|---|---|
| | $\varepsilon_{ATE}$ (TRAIN+VAL) | $\varepsilon_{ATE}$ (TEST) | $\varepsilon_{ATE}$ (TRAIN+VAL) | $\varepsilon_{ATE}$ (TEST) | $\varepsilon_{ATE}$ (TRAIN+VAL) | $\varepsilon_{ATE}$ (TEST) |
| 5 | 0.0533 (0.0165) | 0.0663 (0.0244) | 0.0421 (0.0045) | 0.0472 (0.0166) | 21.0% | 28.8% |
| 10 | 0.0381 (0.0122) | 0.0425 (0.0148) | 0.0296 (0.0040) | 0.0334 (0.0052) | 22.5% | 21.5% |
| 15 | 0.0465 (0.0062) | 0.0545 (0.0112) | 0.0350 (0.0066) | 0.0408 (0.0011) | 24.7% | 25.1% |
| 20 | 0.0764 (0.0178) | 0.0738 (0.0164) | 0.0407 (0.0087) | 0.0383 (0.0054) | 46.6% | 48.1% |

Table 11: Comparison of DMSE with CEVAE on one modality

| MODEL | ATE ERROR (TRAIN+VAL) | ATE ERROR (TEST) |
|---|---|---|
| CEVAE | 0.0637 (0.0162) | 0.0641 (0.0178) |
| DMSE | 0.0328 (0.0040) | 0.0333 (0.0045) |

### H.5.1 Experimental Results

**DMSE under alternative graph structures and comparison with CEVAE:**We generate synthetic data corresponding to Datasets A, B, C and D. Table 13 demonstrates that DMSE recovers ATE under modified causal graph structures. We demonstrate that with increasing dataset size, the ATE error on test dataset continues to fall. Thus, with dataset size approaching infinity, we can recover the true ATE as long as the model class contains true distribution and our optimizer can find the minimum. In Table 12, DMSE also compares favorably with CEVAE and recovers the ATE in this extended setting. In this extended setting, we have a combination of structured and unstructured input modalities. CEVAE takes a concatenation of these modalities as its input while DMSE has a separate model and inference network for each modality. Hence DMSE can handle diverse modality types gracefully as compared to CEVAE.

Table 12: Comparison of CEVAE and DMSE under alternative graph structures

| CAUSAL GRAPH | CEVAE | | DMSE | |
| --- | --- | --- | --- | --- |
| | ATE ERROR (TRAIN+VAL) | ATE ERROR (TEST) | ATE ERROR (TRAIN+VAL) | ATE ERROR (TEST) |
| DATASET A:
ORIGINAL CAUSAL GRAPH | 0.0636 (0.0244) | 0.0752 (0.0276) | 0.0335 (0.0127) | 0.0303 (0.0140) |
| DATASET B:
SOME INPUTS ARE OBSERVED CONFOUNDERS | 0.0522 (0.0134) | 0.0498 (0.0148) | 0.0152 (0.0038) | 0.0237 (0.0049) |
| DATASET C:
SOME INPUTS ARE OBSERVED CONFOUNDERS | 0.0591 (0.0145) | 0.0671 (0.0180) | 0.0315 (0.0055) | 0.0328 (0.0113) |
| DATASET D:
SOME INPUT PROXIES ARE
NOT CONDITIONALLY INDEPENDENT | 0.0375 (0.0141) | 0.0539 (0.0075) | 0.0096 (0.0022) | 0.029 (0.0058) |

Table 13: DMSE under alternative graph structures

| CAUSAL GRAPH | TRAINING DATASET SIZE | ATE ERROR (TRAIN+VAL) | ATE ERROR (TEST) |
| --- | --- | --- | --- |
| DATASET A:
ORIGINAL CAUSAL GRAPH | 100
1000
10000
25000 | 0.1966 (0.0389)
0.0575 (0.0103)
0.0335 (0.0127)
0.0274 (0.0055) | 0.3590 (0.0564)
0.1155 (0.0220)
0.0303 (0.0140)
0.0292 (0.0058) |
| DATASET B:
SOME INPUTS
ARE OBSERVED CONFOUNDERS | 100
1000
10000
25000 | 0.1280 (0.0184)
0.0320 (0.0099)
0.0152 (0.0038)
0.0195 (0.0058) | 0.2770 (0.0401)
0.0951 (0.0264)
0.0237 (0.0049)
0.0204 (0.0063) |
| DATASET C:
SOME INPUTS
ARE OBSERVED CONFOUNDERS | 100
1000
10000
25000 | 0.1354 (0.0361)
0.0596 (0.0191)
0.0315 (0.0055)
0.0140 (0.0039) | 0.1970 (0.0561)
0.1060 (0.0214)
0.0328 (0.0113)
0.0226 (0.0072) |
| DATASET D:
SOME INPUT PROXIES
ARE NOT CONDITIONALLY
INDEPENDENT | 100
1000
10000
25000 | 0.1391 (0.0209)
0.0596 (0.0103)
0.0096 (0.0022)
0.0139 (0.0033) | 0.2300 (0.0544)
0.1012 (0.0223)
0.0290 (0.0058)
0.0206 (0.0045) |

**DMSE under increasing number of modalities and comparison with CEVAE**: We generate synthetic data corresponding to Dataset E with varying number of modalities (i.e. proxies). Table 11 contains the results of experiment with just one modality; the key difference between the two models is the inference procedure. Table 11 shows that even in this setting, DMSE recovers the ATE more accurately than CEVAE. Next, we compare CEVAE vs. DMSE when the data has many unstructured modalities in Table 10. We generate synthetic data from K modalities (Dataset E). The CEVAE model treats them as one concatenated vector; DMSE models them as separate vectors. As expected, DMSE handles large numbers of modalities better than CEVAE.

# I  Identifiability of Causal Effects in the Presence of Proxies to Hidden Confounder

If the structural equations have linear dependencies, we can establish the identifiability of total treatment effect. Kuroki & Pearl [22] define the total treatment effect of $X$ on $Y$ as 'the total sum of the products of the path coefficients on the sequence of arrows along all directed paths from X to Y'. We extend the identifiability result of Kuroki & Pearl [22] to handle vector-valued confounders $\bar{U}$, and hence we require one additional view of the confounder for a total of three independent views. Concretely, we consider the following setup.

$\bar{U}$: hidden confounder

$X$: binary treatment variable

$Y$: univariate outcome variable

$\bar{W}, \bar{Z}, \bar{V}$: proxies for $\bar{U}$

Assume causal graph as

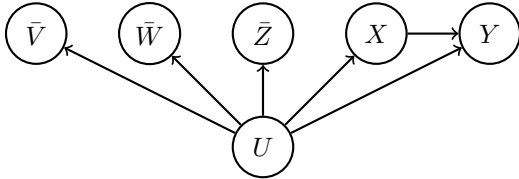

Figure 4: Causal graph for Theorem 4.

Assume linear structural equations

$\bar{W} = \beta_{\bar{W}\bar{U}}\bar{U} + \bar{\epsilon}_W$

$\bar{Z} = \beta_{\bar{Z}\bar{U}}\bar{U} + \bar{\epsilon}_Z$

$\bar{V} = \beta_{\bar{V}\bar{U}}\bar{U} + \bar{\epsilon}_V$

$X = \beta_{X\bar{U}}\bar{U} + \epsilon_X$

$Y = \beta_{Y\bar{U}}\bar{U} + \beta_{YX}X + \epsilon_Y$

Note that the three views of $\bar{U}$ are independent given $\bar{U}$, i.e. $\bar{W} \perp \bar{Z} \perp \bar{V}|\bar{U}$.

**Theorem 4.** *Given the above setup, the causal effect of $X$ on $Y$, i.e. $P(y|do(x))$, is identifiable if the dependency matrices $\beta_{\diamond\bar{U}}$ for $\diamond \in \{\bar{V}, \bar{W}, \bar{Z}\}$ have rank $|\bar{U}|$.*

*Proof Sketch* First, we analyze what happens when we know the $\beta$ matrices a priori, i.e. treatment effect estimation with external information. Then, we analyze how to leverage the three independent views of the confounder to recover enough information about the matrices for treatment effect estimation, i.e. treatment effect estimation without external information. In this second step, we use the rank constraint so that each proxy has enough information about the confounder.

**Treatment effect estimation with external information**:

If any one $\beta$ matrix corresponding to a proxy is known (i.e. we know the causal mechanism $\bar{U} \to \bar{W}$, $\bar{U} \to \bar{Z}$ or $\bar{U} \to \bar{V}$), then the treatment effect is identifiable given that the $\beta$ matrix is invertible. In principle, we need any one proxy and the $\beta$ matrix corresponding to it for computing the treatment effect. The following derivation is done using the matrix $\beta_{\bar{W}\bar{U}}$.

We define $\sigma_{AB} = \text{Covariance}(A, B)$ where $A$ and $B$ are univariate. We define $\Sigma_{\bar{A}\bar{B}}$ to be the covariance matrix between sets of variables in the vectors $\bar{A}$ and $\bar{B}$.

For the above causal graph Figure 4 with linear dependencies, the total treatment effect $\tau_{YX}$ of variable $X$ on output $Y$ can be derived using backdoor adjustment [22] as

$\tau_{YX} = \frac{\sigma_{XY} - \Sigma_{X\bar{U}}\Sigma_{\bar{U}\bar{U}}^{-1}\Sigma_{Y\bar{U}}^T}{\sigma_{XX} - \Sigma_{X\bar{U}}\Sigma_{\bar{U}\bar{U}}^{-1}\Sigma_{X\bar{U}}^T}.$

Using formula of covariance and owing to the linear nature of structural equations, we know that

$\Sigma_{X\bar{W}} = \Sigma_{X\bar{U}}\beta_{\bar{W}\bar{U}}^T, \Sigma_{Y\bar{W}} = \Sigma_{Y\bar{U}}\beta_{\bar{W}\bar{U}}^T.$

Thus, if the matrix $\beta_{\bar{W}\bar{U}}$ is a square and invertible, we get

$\tau_{YX} = \frac{\sigma_{XY} - \Sigma_{X\bar{W}}(\beta_{\bar{W}\bar{U}}\Sigma_{\bar{U}\bar{U}}\beta_{\bar{W}\bar{U}}^T)^{-1}\Sigma_{Y\bar{W}}^T}{\sigma_{XX} - \Sigma_{X\bar{W}}(\beta_{\bar{W}\bar{U}}\Sigma_{\bar{U}\bar{U}}\beta_{\bar{W}\bar{U}}^T)^{-1}\Sigma_{X\bar{W}}^T}.$

**Note**: If the matrix $\beta_{\bar{W}\bar{U}}$ is not square, it needs to have rank $|\bar{U}|$ at the least. In case $|\bar{W}| > |\bar{U}|$, but rank$(\beta_{\bar{W}\bar{U}}) = |\bar{U}|$, we can still work with a modified system of equations such that $\bar{W}$ is replaced with $\bar{W}'$ where $|\bar{W}'| = |\bar{U}| = \text{rank}(\beta_{\bar{W}'\bar{U}})$. To obtain this modified system, we apply elementary row transformations on the equation $\bar{W} = \beta_{\bar{W}\bar{U}}\bar{U} + \bar{\epsilon}_W$ such that $\beta_{\bar{W}\bar{U}}$ is an upper triangular matrix. After that, we drop the bottom $|\bar{W}| - |\bar{U}|$ rows from matrices on both sides of equation to obtain $\bar{W}'$ and $\beta_{\bar{W}'\bar{U}}$.

Consider the case where $\text{rank}(\beta_{\bar{W}\bar{U}}) < |\bar{U}|$ and $\text{rank}(\beta_{\bar{Z}\bar{U}}) < |\bar{U}|$. In this case, **if** we can concatenate the two proxies $\bar{W}$ and $\bar{Z}$ to a new proxy $\bar{V} = \bar{W} : \bar{Z}$ such that $\text{rank}(\beta_{\bar{V}\bar{U}}) = |\bar{U}|$, we can estimate treatment effects using $\beta_{\bar{V}\bar{U}}$.

**Treatment effect estimation without external information**:

In this case, we need three independent views of hidden confounder U. As we are working with multivariate confounders, the univariate treatment variable X cannot serve as the third view of U anymore.

Now using the properties of covariance matrix and the linearity of structural equations, we can write the following

$$\Sigma_{\bar{V}\bar{W}} = \beta_{\bar{V}\bar{U}}\Sigma_{\bar{U}\bar{U}}\beta_{\bar{W}\bar{U}}^T, \Sigma_{\bar{W}\bar{Z}} = \beta_{\bar{W}\bar{U}}\Sigma_{\bar{U}\bar{U}}\beta_{\bar{Z}\bar{U}}^T, \Sigma_{\bar{V}\bar{Z}} = \beta_{\bar{V}\bar{U}}\Sigma_{\bar{U}\bar{U}}\beta_{\bar{Z}\bar{U}}^T.$$

Assuming that the matrices $\Sigma_{\bar{U}\bar{U}}, \beta_{\bar{V}\bar{U}}, \beta_{\bar{Z}\bar{U}}$ are square and invertible, we get $\Sigma_{\bar{W}\bar{Z}}\Sigma_{\bar{V}\bar{Z}}^{-1}\Sigma_{\bar{V}\bar{W}} = \beta_{\bar{W}\bar{U}}\Sigma_{\bar{U}\bar{U}}\beta_{\bar{W}\bar{U}}^T$

Hence, if $\beta_{\bar{W}\bar{U}}$ is invertible additionally, then the treatment effect can be identified using the following equation

$$\tau_{YX} = \frac{\sigma_{XY} - \Sigma_{X\bar{W}}(\Sigma_{\bar{W}\bar{Z}}\Sigma_{\bar{V}\bar{Z}}^{-1}\Sigma_{\bar{V}\bar{W}})^{-1}\Sigma_{Y\bar{W}}^T}{\sigma_{XX} - \Sigma_{X\bar{W}}(\Sigma_{\bar{W}\bar{Z}}\Sigma_{\bar{V}\bar{Z}}^{-1}\Sigma_{\bar{V}\bar{W}})^{-1}\Sigma_{X\bar{W}}^T}$$

∎

**What happens when the dimensionality of proxies is greater than dimensionality of true confounder?**

We are interested in high-dimensional, unstructured proxies. In this case, our matrices $\beta_{\bar{V}\bar{U}}, \beta_{\bar{Z}\bar{U}}, \beta_{\bar{W}\bar{U}}$ will need to have rank $|\bar{U}|$. In essence, we still need three views for our vector $\bar{U}$. When we have access to external information in the form of $\beta$ matrices, verifying the rank is possible. We can then apply appropriate row transformations on the corresponding structural equations (as mentioned in I) to obtain modified proxies with same length as the confounder $\bar{U}$.

In absence of external information, rank of $\beta$ matrices is not verifiable. Assuming that we have $\beta_{\bar{V}\bar{U}}, \beta_{\bar{Z}\bar{U}}, \beta_{\bar{W}\bar{U}}$ with rank $|\bar{U}|$, we apply a dimensionality reduction procedure to map the proxies $\bar{V}, \bar{Z}, \bar{W}$ to modified proxies $\bar{V}', \bar{Z}', \bar{W}'$ such that $|\bar{V}'| = |\bar{Z}'| = |\bar{W}'| = |\bar{U}|$ and $\beta_{\bar{V}'\bar{U}}, \beta_{\bar{Z}'\bar{U}}, \beta_{\bar{W}'\bar{U}}$ have rank $|\bar{U}|$. In practice, this dimensionality reduction can be a technique like PCA or the result of applying pre-trained neural network layer to obtain an embedding of unstructured data.

## J   Additional Details on Mathematical Proofs

### J.1   Evidence Lower Bound for Deep Structural Equations

We discuss the evidence lower bound (ELBO), presented in Section 4.2.

The left-hand-side of the ELBO can be written as

$$\sum_{i=1}^{n}\mathbb{E}_q\left[\sum_{j=1}^{m}\log p(x_i^j|z) + \log p(y_i|z) + \log p(y_i, t_i, z) - \log q(z|x_i, t_i, y_i)\right]. \tag{10}$$

Due to the causal graph structure we can factorize the following distribution as

$$p(y_i, t_i, z) = \log p(y_i|t_i, z) + \log p(t_i|z) + \log p(z). \tag{11}$$

Thus, we can rewrite (10) as

$$\sum_{i=1}^{n}\mathbb{E}_q\left[\sum_{j=1}^{m}\log p(x_i^j|z) + \log p(y_i|z) + \log p(y_i|t_i, z) + \log p(t_i|z) + \log p(z) - \log q(z|x_i, t_i, y_i)\right] \tag{12}$$

Note that the first four terms in the expectation over the posterior distribution $q$ make up a reconstruction loss of the original data $x, y, t$ in terms of the latent variable $z$, while the last two terms form

the negative KL divergence $-D_{KL}\left(q(z|x_i, t_i, y_i)\|p(z)\right)$ between the posterior $q$ and the prior $p(z)$. Thus, we can write our ELBO as

$$\sum_{i=1}^{n} \mathcal{L}_{\text{reconstruction}}(x_i, y_i, t_i, z) - D_{KL}\left(q(z|x_i, t_i, y_i)\|p(z)\right), \tag{13}$$

which is exactly our variational objective. We can sum or take the average over the number of datapoints $n$ to form our empirical objective.

## J.2 Derivation of posterior

Here we show why the true posterior factorizes as

$$p(z \mid x, t, y) \propto (p(z \mid t, y) \prod_{j=1}^{m} p(z \mid x_j))/ \prod_{j=1}^{m-1} p(z).$$

$$
\begin{aligned}
p(z \mid x, t, y) &= p(x, t, y, z)/p(x, t, y) \\
&= \left(\prod_{j=1}^{m} p(x_j \mid z)\right) p(t, y \mid z)p(z)/p(x, t, y) \quad \text{(by cond. indep. of causal graph)} \\
&= \left(\prod_{j=1}^{m} p(z \mid x_j)p(x_j)/p(z)\right) \cdot (p(z \mid t, y)p(t, y)p(z)/p(x, t, y)) \\
&\propto \left(\prod_{j=1}^{m} p(z \mid x_j)/p(z)\right) \cdot (p(z \mid t, y)p(z)). \quad \text{(removing terms independent of } z\text{)}
\end{aligned}
$$