# OpenReview forum: "Deep Multi-Modal Structural Equations For Causal Effect Estimation With Unstructured Proxies"
_NeurIPS.cc/2022/Conference — NeurIPS 2022 Accept_

### Official Review · Reviewer_zkCX · 2022-06-23

**Rating:** 5
**Confidence:** 3
**Soundness:** 2 fair
**Presentation:** 3 good
**Contribution:** 2 fair

**Summary:**

This work addresses the problem of estimating the unbiased causal effect of an intervention on the outcome using unstructured proxy variables. The authors first formulate the underlying causal structure of the problem. The authors then develop a generative approach, i.e., estimating the parameters of the deep multi-modal structural equations (DMSEs) or Deep Gaussian Structural Equations (DGSEs) via optimizing the derived multi-model evidence lower bound. The causal effects, therefore, can be computed from the learned models. The experiments used real-world datasets (IHDP, STAR, and GWAS datasets) to demonstrate that the developed models can estimate ATE with fewer errors than competing methods.

**Questions:**


This work heavily relies on the assumption that the causal structured is the form $$ X = f_1(Z, \epsilon_1), T = f_2(Z, \epsilon_2), Y = f_3(Z, Y, \epsilon_3). $$  The causal effect estimation from the DMSE or DGSE is unbiased when the causal structure is indeed true. However, many other dependencies between these variables make sense in the real-world setting (such as the existence of the interdependence between proxy variables or the proxy variables also control the distribution of the intervention variable and the outcome variable), which the author did not discuss.

The following questions should be discussed:

Why is only the causal structure shown in Figure 1 discussed?
What are the relevant issues, such as whether it is possible to model it using a generative model?
Is it able to estimate the ATE from it?
What might go wrong if the underlying model is inconsistent?

Also, the following paper might be helpful for the discussion:

Miao, Wang, Zhi Geng, and Eric J. Tchetgen Tchetgen. "Identifying causal effects with proxy variables of an unmeasured confounder." Biometrika 105, no. 4 (2018): 987-993.

**Limitations:**

Similar to the above discussion, it is beneficial that the authors could discuss the assumption made on the causal model and the possible limitations or societal impact on the real-world applications.

**Strengths And Weaknesses:**


Strengths:
1) The authors provide motivating real-world applications (Healthcare and Genomics) to show the context where a large amount of unstructured data is available and the missing confounders' issues in these respective applications.
2) The notations, problem formulation, the underlying causal model, the derivation of the objective, and the estimation of causal effects are presented.
3) Details of the experimental setup are well described in the manuscript and the appendix.

Weaknesses:
1) The central issue of this work heavily relies on the causal structure they are considering (shown in Figure 1), which I will discuss in more detail in the next section).
2) In the experimental section if the STAR dataset were used for the experimental validation, its results (i.e., Table 6 in the appendix) should also be presented in the manuscript to help readers read smoothly.

---

> ### Author Response · Authors · 2022-08-03
> **On Causal Graph Assumptions and Novelty**
>
> We thank the reviewer for detailed comments and feedback on our paper.
>
> **On Causal Graph Assumptions**:
>
> The reviewer’s statement can also be interpreted more broadly as a concern that all $X_i$ are conditionally independent proxy variables. To address this potential concern, **we derive new extensions to our methods** to the following settings:
>
> - In addition to proxy variables $X_i$, we also observe a covariate $V$ that represents observed confounders (i.e., we always have it in the data, and it influences $Y$, $T$).
> - The proxy variables may have mutual dependencies, i.e. **$X_i$** and **$X_j$** may be connected by edges in the causal graph.
>
> We formally define these extensions in a new **Appendix G**, and we also derive the following results:
>
> - The true causal effect of **$T$** is identifiable in the more general setting presented above (**Theorem 2**).
> - We propose a new model family, **DMSE-V**, which extends **DMSE** to the above setting via simple modifications (e.g., the introduction of a new variable).
> - We derive learning and inference algorithms for **DMSE-V**. The result of these algorithms is an estimator for the ATE and ITE that holds in the extended setting.
>     - The learning and inference algorithms are minor modifications to our existing ones. In particular, they involve adding the extra variable $V$ as input to the distributions $P(Y | T, Z, V), P(T | Z, V), Q(Z | Y, T, V, X)$, and the learning algorithms remain the same.
>
> For full details, please see **Appendix G**.
>
> **On comparing our methods with CEVAE/other VAE based estimators**:
>
> The reviewer is concerned about the novelty of our method relative to previous estimators of ATE/ITE based on deep structural equations and generative models (e.g., CEVAE).
>
> While our method is an instance of generative models, we identify the following key differences:
>
> - We propose **new generative model architectures** that extend existing models (e.g., DSE, CEVAE) to multiple proxies $X_i$, each possibly coming from a different modality.
> - We derive **novel inference algorithms** for these extended models, which have the following benefits:
>     - Our algorithms scale better to large sets of modalities by leveraging the independence structure of the $X_i$.
>     - Our inference algorithms naturally handle missing $X_i$.
>     - They are also simpler: they don’t require auxiliary networks (e.g., like in CEVAE).
> - Lastly, our key contribution is that we demonstrate the effectiveness of generative models at **modeling unstructured proxies** (many previous methods instead relied on propensity scoring).

---

> > ### Author Response · Authors · 2022-08-03
> > **Discussion of Reviewer Questions About Model Structure**
> >
> > **Discussion of Reviewer Questions About Model Structure**
> >
> > *Why is only the causal structure shown in Figure 1 discussed?*
> >
> > We first note that Figure 1 describes a sensible structure for our setting: unstructured data are most likely proxies and are unlikely to influence other variables (e.g., pixel values are unlikely to have a direct causal effect on either X, Y, or T). The proxies can also be structured in our framework (e.g., in our IHDP experiments).
> >
> > However, we acknowledge that other structures are possible. As discussed above, we have introduced extensions to when there are dependencies among the proxies, and some covariates are fully observed (**Appendix G**). These are simple extensions of our framework in Figure 1.
> >
> > *What are the relevant issues, such as whether it is possible to model it using a generative model?*
> >
> > Assuming the causal structure holds, we may fit a generative model in order to recover the data distribution. There are several considerations:
> >
> > - *Does the causal structure yield identifiable causal effects?* We prove that our structure does (Theorem 1).
> >
> > - *Are we able to fit the model to the data distribution?* We take an approach of fitting an expressive neural estimator that can approximate the data distribution well.
> >
> > - *Can we obtain an estimate of ITE from the model once we learn it?* We derive an expression for the ITE as a function of the estimated model and provide approximate variational inference algorithms (Section 4) for calculating that expression.
> >
> > Some previous works (e.g., Miao et al.) fit a generative model with a simple (e.g., linear) parameterization. This yields guarantees on being able to recover the true data distribution if it also has that simple parameterization. However, in practice, the latter assumption is rarely true. An alternative approach is to use a very flexible neural estimator. It is not guaranteed to recover the true distribution in theory, but in practice, it tends to fit the data distribution better than simpler models.
> >
> > *Is it able to estimate the ATE from it? What might go wrong if the underlying model is inconsistent?*
> > Yes, we can estimate the ATE from the model using the procedure applied above. However, as we just mentioned this assumes that (1) our model class includes the true data distribution; (2) we have enough data to learn the model in our class; (3) our learning algorithm (objective function optimizer) can identify the true model (assuming we had infinite data); (4) we can compute ITE estimates from the model. Failure modes (1) and (2) hold for any generative models; (3) and (4) are more specific to deep generative models. In practice, we can attempt to verify if failure modes (1), (2), (3) occurred by evaluating the generative model on a hold-out dataset (and we identify these failures given enough data). We can also try to mitigate failure (4) by using a more sophisticated inference algorithm (e.g., MCMC), but that remains a direction for future work.
> >
> > **Discussion of Paper by Miao et al.**
> >
> > Our work is of a similar flavor to that of Miao et al., with two key differences: Miao et al. study the setting of simple (non-neural) dependencies among variables. Our work can be seen as extending their model to neural parameterizations. Since we use neural models, we cannot provide rigorous guarantees (e.g., we don’t know if the neural network will converge); however, our neural models can incorporate unstructured data, leading to improved performance.
> >
> >
> >
> >
> > **STAR Experiment Table**:
> > Thank you for pointing out that having this table in the main text might be more useful. We moved the Table 6 on STAR experiments to the appendix due to space limitations since the IHDP experimental table demonstrated similar results. We can move this table to the main paper so that it is easier to connect the experimental results together.
> >
> >
> >
> >
> > Citations:
> >
> > [1] Pearl, J. Causality. Cambridge University Press, 2009

---

> > > ### Comment · Reviewer_zkCX · 2022-08-08
> > > **Thank you Authors for the Feedback**
> > >
> > > I sincerely thank the authors for their significant effort. I appreciate the extensions and derivations of the current model in Appendix G.
> > > I thank the authors for discussing the possibility of other causal structures and how to identify the inconsistency of the model using the hold-out dataset. I would like to see a discussion or a simulation in the updated version. I maintain my current score.

---

> > > > ### Author Response · Authors · 2022-08-09
> > > > **Additional Simulation Experiments**
> > > >
> > > > We thank the reviewer for their response. The reviewer would like to see additional experiments and simulations to support our claims. Below, we report __additional experiments__ on two settings: modified causal graph structures and a head-to-head comparison of DMSE vs. CEVAE.
> > > >
> > > > **Experiments with Modified Causal Graph Structures**
> > > >
> > > > In our previous response, we argued that our core method handles modified causal graph structures with possibly small modifications. Here, we perform a simulation study to confirm these results empirically. In our experiments, we find that DMSE __recovers the ATE with similar accuracy on the modified and unmodified graph structures__ and its __error tends to zero with increasing dataset sizes__.
> > > >
> > > > Below, we use an extension of the synthetic dataset used in our paper: see **Appendix G.5** for the full data generating process. Table 1 below demonstrates that DMSE recovers ATE under modified causal graph structures. We demonstrate that with increasing dataset size, the ATE error on test dataset continues to fall. Thus, with dataset size approaching infinity, we can recover the true ATE as long as the model class contains true distribution and our optimizer can find the minimum.
> > > >
> > > > **Table 1: DMSE under alternative graph structures**
> > > > | Causal Graph | Training Dataset Size | ATE error (train+val) |ATE error (test)
> > > > |--- |--- |--- |---|
> > > > |Dataset A: Original causal graph|100     | 0.1966 (0.0389)  | 0.3590 (0.0564)
> > > > || 1000  | 0.0575 (0.0103)  | 0.1155 (0.0220)
> > > > || 10000 | 0.0335 (0.0127) | 0.0303 (0.0140)
> > > > || 25000 | 0.0274 (0.0055) | 0.0292 (0.0058)
> > > > Dataset B: Some inputs are observed confounders|100     | 0.1280  (0.0184)  | 0.2770 (0.0401)
> > > > || 1000  | 0.0320 (0.0099)  | 0.0951 (0.0264)
> > > > || 10000 | 0.0152 (0.0038) | 0.0237 (0.0049)
> > > > || 25000 | 0.01951 (0.0058) | 0.0204 (0.0063)
> > > > |Dataset C: Some inputs are observed confounders|100     | 0.1354 (0.0361)  | 0.1970 (0.0561)
> > > > || 1000  |0.0596 (0.0191)  | 0.1060 (0.0214)
> > > > || 10000 | 0.0315 (0.0055) | 0.0328 (0.0113)
> > > > || 25000 | 0.0140 (0.0039) | 0.0226 (0.0072)
> > > > |Dataset D: Some input proxies are not conditionally independent|100     | 0.1391 (2.09E-02)  | 0.2300 (5.44E-02)
> > > > || 1000  |0.0596 (0.0103)  | 0.1012 (0.0223)
> > > > || 10000 | 0.0096 (0.0022) | 0.029 (0.0058)
> > > > || 25000 | 0.0139 (0.0033) | 0.0206 (0.0045)
> > > >
> > > > In Table 2, DMSE also compares favorably with CEVAE and recovers the ATE in this extended setting. In this extended setting, we have a combination of structured and unstructured input modalities. CEVAE takes a concatenation of these modalities as its input while DMSE has a separate model and inference network for each modality. Hence DMSE can handle diverse modality types gracefully as compared to CEVAE.
> > > >
> > > > Thus, in response to questions from reviewers **5MY4** and **zkCX**, we empirically demonstrate that our methods can work with modified causal graph structures (all input covariates do not need to be conditionally independent unstructured proxies). This supports the discussion we have newly added in **Appendix G** during the rebuttal period.
> > > >
> > > > **Table 2: Comparison of CEVAE and DMSE under alternative graph structures**
> > > > | Causal Graph | CEVAE | | DMSE | |
> > > > | --- | --- | --- | --- | --- |
> > > > | | ATE error (train+val) |ATE error (test) | ATE error (train+val) | ATE error (test)|
> > > > | Dataset A: Original causal graph |0.0636 (0.0244) | 0.0752 (0.0276) | 0.0335 (0.0127) | 0.0303 (0.0140) |
> > > > | Dataset B: Some inputs are observed confounders | 0.0522 (0.0134) | 0.0498 (0.0148) | 0.0152 (0.0038) | 0.0237 (0.0049) |
> > > > | Dataset C: Some inputs are observed confounders | 0.0591 (0.0145) | 0.0671 (0.0180) |0.0315 (0.0055) | 0.0328 (0.0113) |
> > > > | Dataset D: Some input proxies are not conditionally independent | 0.0375 (0.0141) | 0.0539 (0.0075) | 0.0096 (0.0022) | 0.029 (0.0058) |

---

> > > > > ### Author Response · Authors · 2022-08-09
> > > > > **Additional Simulation Experiments**
> > > > >
> > > > > **Comparing of DMSE with CEVAE**
> > > > >
> > > > > In our previous response, we identified key differences between DMSE and CEVAE: (1) a modified causal graph architecture that supports multiple unstructured and possibly missing proxies; (2) improved inference algorithms. We now show that these differences lead to differences in performance in practice by performing a simulated experiment. In all settings, we **outperform the popular CEVAE model**.
> > > > >
> > > > > First, we generate synthetic data using a process similar to the synthetic experiment already present in our paper (see Appendix G.5 for details). There is only one modality in this experiment; the key difference between the two models in the inference procedure. Table 1 shows that even in this setting, DMSE recovers the ATE more accurately than CEVAE.
> > > > >
> > > > > **Table 1: Comparison of DMSE with CEVAE on one modality**
> > > > > | Model | ATE error (train+val) | ATE error (test) |
> > > > > | --- | --- | --- |
> > > > > | CEVAE | 6.37E-02  (1.62E-02) | 6.41E-02 (1.78E-02) |
> > > > > | DMSE | 3.28E-02 (3.96E-03) | 3.33E-02 (4.49E-03) |
> > > > >
> > > > > Next, we compare CEVAE vs. DMSE when the data has **many unstructured modalities** in Table 2. We generate synthetic data from K modalities. The details on this are in **(Appendix G.5) (Dataset E)**. The CEVAE model treats them as one concatenated vector; DMSE models them as separate vectors. As expected, DMSE handles large numbers of modalities better than CEVAE.
> > > > >
> > > > > **Table 2: Comparison of DMSE with CEVAE under increasing number of input proxies**
> > > > > | Number of input modalities | CEVAE | | DMSE | | % improvement made by DMSE w.r.t CEVAE | |
> > > > > | --- | --- | --- | --- | --- | --- | --- |
> > > > > | | ATE error (train+val) | ATE error (test) | ATE error (train+val) | ATE error (test) | ATE error (train+val) | ATE error (test) |
> > > > > | 5 | 0.0533 (0.0165) | 0.0663 (0.0244) | 0.0421 (0.0045) | 0.0472 (0.0166) | 21.0% | 28.8% |
> > > > > | 10 | 0.0381 (0.0122) | 0.0425 (0.0148) | 0.0296 (0.0040) | 0.0334 (0.0052) | 22.5% | 21.5% |
> > > > > | 15 | 0.0465 (0.0062) | 0.0545 (0.0112) | 0.0350 (0.0066) | 0.0408 (0.0011) | 24.7% | 25.1% |
> > > > > | 20 | 0.0764 (0.0178) | 0.0738 (0.0164) | 0.0407 (0.0087) | 0.0383 (0.0054) | 46.6% | 48.1% |

---

> > > > > > ### Author Response · Authors · 2022-08-09
> > > > > > **Additional Experiments**
> > > > > >
> > > > > > **Experiments with Modified Causal Graph Structures**
> > > > > >
> > > > > > In our previous response, we argued that our core method handles modified causal graph structures with possibly small modifications. Here, we perform a simulation study to confirm these results empirically. In our experiments, we find that DMSE __recovers the ATE with similar accuracy on the modified and unmodified graph structures__ and its __error tends to zero with increasing dataset sizes__.
> > > > > >
> > > > > > Below, we use an extension of the synthetic dataset used in our paper: see **Appendix G.5** for the full data generating process. Table 1 below demonstrates that DMSE recovers ATE under modified causal graph structures. We demonstrate that with increasing dataset size, the ATE error on test dataset continues to fall. Thus, with dataset size approaching infinity, we can recover the true ATE as long as the model class contains true distribution and our optimizer can find the minimum.
> > > > > >
> > > > > > **Table 1: DMSE under alternative graph structures**
> > > > > > | Causal Graph | Training Dataset Size | ATE error (train+val) |ATE error (test)
> > > > > > |--- |--- |--- |---|
> > > > > > |Dataset A: Original causal graph|100     | 0.1966 (0.0389)  | 0.3590 (0.0564)
> > > > > > || 1000  | 0.0575 (0.0103)  | 0.1155 (0.0220)
> > > > > > || 10000 | 0.0335 (0.0127) | 0.0303 (0.0140)
> > > > > > || 25000 | 0.0274 (0.0055) | 0.0292 (0.0058)
> > > > > > Dataset B: Some inputs are observed confounders|100     | 0.1280  (0.0184)  | 0.2770 (0.0401)
> > > > > > || 1000  | 0.0320 (0.0099)  | 0.0951 (0.0264)
> > > > > > || 10000 | 0.0152 (0.0038) | 0.0237 (0.0049)
> > > > > > || 25000 | 0.01951 (0.0058) | 0.0204 (0.0063)
> > > > > > |Dataset C: Some inputs are observed confounders|100     | 0.1354 (0.0361)  | 0.1970 (0.0561)
> > > > > > || 1000  |0.0596 (0.0191)  | 0.1060 (0.0214)
> > > > > > || 10000 | 0.0315 (0.0055) | 0.0328 (0.0113)
> > > > > > || 25000 | 0.0140 (0.0039) | 0.0226 (0.0072)
> > > > > > |Dataset D: Some input proxies are not conditionally independent|100     | 0.1391 (2.09E-02)  | 0.2300 (5.44E-02)
> > > > > > || 1000  |0.0596 (0.0103)  | 0.1012 (0.0223)
> > > > > > || 10000 | 0.0096 (0.0022) | 0.029 (0.0058)
> > > > > > || 25000 | 0.0139 (0.0033) | 0.0206 (0.0045)
> > > > > >
> > > > > > In Table 2, DMSE also compares favorably with CEVAE and recovers the ATE in this extended setting. In this extended setting, we have a combination of structured and unstructured input modalities. CEVAE takes a concatenation of these modalities as its input while DMSE has a separate model and inference network for each modality. Hence DMSE can handle diverse modality types gracefully as compared to CEVAE.
> > > > > >
> > > > > > Thus, in response to questions from reviewers **5MY4** and **zkCX**, we empirically demonstrate that our methods can work with modified causal graph structures (all input covariates do not need to be conditionally independent unstructured proxies). This supports the discussion we have newly added in **Appendix G** during the rebuttal period.
> > > > > >
> > > > > > **Table 2: Comparison of CEVAE and DMSE under alternative graph structures**
> > > > > > | Causal Graph | CEVAE | | DMSE | |
> > > > > > | --- | --- | --- | --- | --- |
> > > > > > | | ATE error (train+val) |ATE error (test) | ATE error (train+val) | ATE error (test)|
> > > > > > | Dataset A: Original causal graph |0.0636 (0.0244) | 0.0752 (0.0276) | 0.0335 (0.0127) | 0.0303 (0.0140) |
> > > > > > | Dataset B: Some inputs are observed confounders | 0.0522 (0.0134) | 0.0498 (0.0148) | 0.0152 (0.0038) | 0.0237 (0.0049) |
> > > > > > | Dataset C: Some inputs are observed confounders | 0.0591 (0.0145) | 0.0671 (0.0180) |0.0315 (0.0055) | 0.0328 (0.0113) |
> > > > > > | Dataset D: Some input proxies are not conditionally independent | 0.0375 (0.0141) | 0.0539 (0.0075) | 0.0096 (0.0022) | 0.029 (0.0058) |

---

### Official Review · Reviewer_GQXK · 2022-07-05

**Rating:** 7
**Confidence:** 4
**Soundness:** 3 good
**Presentation:** 3 good
**Contribution:** 3 good

**Summary:**

This paper addresses causal effect estimation when there are unobserved confounders but there exist (often unused) unconstrained data that could be used as proxies for the unobserved confounders. A model named Deep Multi-modal Structural Equations (DMSE) is proposed to leverage the multi-modal data. The experimental results demonstrate the effectiveness of the proposed method on two real-world tasks.


**Questions:**

- Lines 103-104 state that “our approach uses deep structural equations to extract confounding signal from the multi-modal proxies”. The “deep” part does this not the “structural equations” part, correct? Because SEs cannot do representation learning.
- How did you arrive at Eq. (5)? ELBO should have a reconstruction error part and a KL divergence part. Please provide detail on how you derived this ELBO.
- Lines 137-139 state that the posterior factorizes in that way. Could you please elaborate on why this is the case?
- Line 155: please elaborate why auxiliary inference networks are not necessary in your application?


**Strengths And Weaknesses:**

Pros:
- The paper is written clearly and it’s easy to read.
- The empirical experiments are thorough.
- The contribution is significant in that the proposed model can leverage often unused and sometimes missing multi-modal data to enhance causal effect estimation performance.

Cons:
- The originality of the paper seems a bit limited as the proposed method is composed of already available components (i.e., VAE, SE).
- There is little explanation for the main equations stated in the paper. I will comment on these in the “Questions” section.

---

> ### Author Response · Authors · 2022-08-03
> **On novelty of our techniques and answers to specific questions**
>
> We thank the reviewer for detailed comments and feedback on our paper.
>
> **On The Novelty of Our Techniques**:
>
> The reviewer is concerned about the novelty of our method relative to previous estimators of ATE/ITE based on deep structural equations and generative models (e.g., CEVAE).
>
> Working with multiple unstructured modalities (some of which may  be missing) requires us to develop novel approximate variational inference techniques that improve over existing generative models. Specifically, we identify the following key differences:
>
> - We propose **new generative model architectures** that extend existing models (e.g., DSE, CEVAE) to multiple proxies $X_i$, each possibly coming from a different modality.
> - We derive **novel inference algorithms** for these extended models, which have the following benefits:
>     - Our algorithms scale better to large sets of modalities by leveraging the independence structure of the $X_i$.
>     - Our inference algorithms naturally handle missing $X_i$.
>     - They are also simpler: they don’t require auxiliary networks (e.g., like in CEVAE).
> - Lastly, our key contribution is that we demonstrate the effectiveness of generative models at **modeling unstructured proxies** (many previous methods instead relied on propensity scoring).
>
> - Using modality specific inference networks also allows us to use modality-specific architectures separately, allowing us to process diverse modalities like genomic sequences, images or tabular data at the same time.
>
> **On Additional Questions**
>
> *Clarification on ‘Deep’ Structural Equations*
>
> The reviewer is right in pointing out that linear structural equations by themselves cannot learn representations. For this reason, we use a deep neural network parameterization to extract useful features from unstructured information. For example, we can use neural networks specific to a modality (e.g CNNs) to extract features like age or gender from the image of a person.
>
> *ELBO derivation*:
>
> Our reconstruction term is $E_q \log (p(x, y, t|z))$. We utilize the conditional independencies within the causal graph in Figure 1 to factorize $p(x, y ,t | z)$.
>
> $E_q \log (q(z| x, t, y)/p(z))$ corresponds to the KL divergence term. Please refer to **Appendix H.1** for a detailed derivation.
>
> *On the derivation for factorization of posterior:*
>
> $p(z|x,t,y) ∝ (p(z|t, y) \prod_{j=0}^{j=m}p(z|x_j))/( \prod_{j=0}^{j=m-1} p(z))$
>
> This factorization is derived by using the conditional independencies between the input modalities $x_i$ given hidden confounder $z$ as implied by the causal graph. We apply Bayes rule to obtain conditioning of variables on $z$ and then exploit the conditional independencies to factorize the distribution.  Please refer to **Appendix H.2** for full derivation.
>
> Here we assume that if the true posterior components $p(z|x_i)$ and $p(z|t, y)$ are contained in the variational counterparts $q(z|x_i)$ and $q(z| t, y)$ respectively, then we can obtain the factorization as approximately equal to $q(z|t, y) \prod_{j=0}^{j=m}(q(z| x_j) ) / (\prod_{j=0}^{j=m-1} p(z))$
>
>
> *Why are auxiliary networks not necessary?*
>
> The CEVAE involves using additional auxiliary networks $Q(T|X)$ and $Q(Y|X, T)$ when inferring the posterior $Q(Z|X)$ while estimating treatment effects [1]. Since we use a product-of-experts formulation, we do not need to train these additional networks. We can compute $Q(Z|X)$ from the $Q(Z|X_i)$ networks trained over each individual modality $X_i$ as shown in equation (7) in our paper, which also allows us to handle missing modalities gracefully.
>
> We will modify the descriptions of the above parts so that the equations are easier to follow.
>
> We would also like to point the reviewer to additional discussion in **Appendix G** concerning alternative causal graph structures added in response to other reviewer comments.
>
> [1] Christos Louizos, Uri Shalit, Joris Mooij, David Sontag, Richard Zemel, Max Welling. Causal Effect Inference with Deep Latent-Variable Models. 2017

---

> > ### Comment · Reviewer_GQXK · 2022-08-06
> > **Thank you for the clarifications**
> >
> > The authors have addressed my concerns in their rebuttal.
> > As a result, I have increased my score accordingly.

---

### Official Review · Reviewer_5MY4 · 2022-07-11

**Rating:** 5
**Confidence:** 4
**Soundness:** 2 fair
**Presentation:** 2 fair
**Contribution:** 3 good

**Summary:**

This paper presents novel deep multi-modal structural equations for causal effect estimation from unstructured data with unobserved confounders. The correctness of the developed method relies on the set of rich unstructured proxy variables and perfect modelling. The experiments on synthetic and semi-synthetic datasets show the performance of the developed method.

**Questions:**

Q1. Why not compare the developed methods with CEVAE and other VAE-based estimators in the experiments?

Q2. What are the missing modalities $X_j$ in this work?

**Limitations:**

Yes, I have not found a negative societal impact.

**Strengths And Weaknesses:**

This paper considers a very important problem in causal inference, i.e. estimating the causal effect of intervention from unstructured data with latent confounders. The paper seems to take the main step of CEVAE over some specialized architectures, such as convolutions for images. The idea of the paper is good and interesting. However, to my understanding, the assumption of all covariates are the unstructured proxy variables, seems too strong to be satisfied in many real-world applications. For example, what is the latent confounder of the proxy variable "sex of baby" in terms of IHDP? This may not be very practical. Moreover, there are many issues with the presentations that make them not easy to follow. Overall, the paper is not good enough.

Some minors:

*multiple multi-model, it is better to remove `multiple'.

*Confounder does not define in this paper.

*Are "unstructured proxy variables", "unstructured multi-modal proxy variables" and "unstructured data" the same in this work? It is better to keep consistent.

*Line 143 to 147, It would be better to provide a more detailed explanation. The current evidence and conclusion given are somewhat sloppy.

*After Line 198, "1", "0" -> \``1", \``0"

*The conclusion in Section 5.1 is not clear from the current descriptions.

=== Strengths ===

1. A novel deep multi-modal  structure equation is developed for unstructured data

2. Experiments conducted on a number of datasets show the performance of the developed algorithm.

=== Weaknesses ===

There is not a theoretical analysis of the developed deep multi-modal structure equations.

---

> ### Author Response · Authors · 2022-08-03
> **On Assumptions, Alternate Causal Graph Structures and Theoretical Justification**
>
> We thank the reviewer for detailed comments and feedback on our paper.
>
> **On the assumption that all input covariates are unstructured proxies**:
>
> The reviewer is concerned that our model treats all input covariates as unstructured proxies. First of all, **this is a misunderstanding**: in our experiment, we use both structured and unstructured proxies.
>
> The reviewer’s statement can also be interpreted more broadly as a concern that all $X_i$ are conditionally independent proxy variables. To address this potential concern, we derive new extensions to our methods to the following settings:
> In addition to proxy variables $X_i$, we also observe a covariate $V$ that represents observed confounders (i.e., we always have it in the data, and it influences $Y$, $T$).
> The proxy variables may have mutual dependencies, i.e. **$X_i$** and **$X_j$** may be connected by edges in the causal graph.
>
> We formally define these extensions in a new **Appendix G**, and we also derive the following results:
>
> - The true causal effect of **$T$** is identifiable in the more general setting presented above (**Theorem 2**).
> - We propose a new model family, **DMSE-V**, which extends **DMSE** to the above setting via simple modifications (e.g., the introduction of a new variable).
> - We derive learning and inference algorithms for **DMSE-V**. The result of these algorithms is an estimator for the ATE and ITE that holds in the extended setting.
>     - The learning and inference algorithms are minor modifications to our existing ones. In particular, they involve adding the extra variable $V$ as input to the distributions $P(Y | T, Z, V), P(T | Z, V), Q(Z | Y, T, V, X)$, and the learning algorithms remain the same.
>
> **On comparing our methods with CEVAE/other VAE based estimators**:
>
> The reviewer is concerned about the novelty of our method relative to previous estimators of ATE/ITE based on deep structural equations and generative models (e.g., CEVAE).
>
> While our method is an instance of generative models, we identify the following key differences:
>
> - We propose **new generative model architectures** that extend existing models (e.g., DSE, CEVAE) to multiple proxies $X_i$, each possibly coming from a different modality.
> - We derive **novel inference algorithms** for these extended models, which have the following benefits:
>     - Our algorithms scale better to large sets of modalities by leveraging the independence structure of the $X_i$.
>     - Our inference algorithms naturally handle missing $X_i$.
>     - They are also simpler: they don’t require auxiliary networks (e.g., like in CEVAE).
> - Lastly, our key contribution is that we demonstrate the effectiveness of generative models at **modeling unstructured proxies** (many previous methods instead relied on propensity scoring).
>
> **Theoretical Justifications for The Proposed Method**
>
> Our graphical structure is directly inspired by theoretical results for models with similar structures, but that assume linear (rather than neural) functions between the variables.
>
> Most notably, Kuroki and Pearl give identifiability results for when the latent $Z$ and observed $X_i$’s are discrete, and $X_i$’s are different views of $Z$. The model is identifiable precisely only when there are at least two conditionally independent proxy variables. Our assumption of multiple independent proxies mirrors theirs.
>
> When $Z$ and $X_i$’s are continuous, a natural first step is the linear case: when $X_i$’s are different noisy and potentially missing views of the latent $Z$, and that the outcomes $Y$ are linear in $Z$. Under appropriate regularity conditions, Kallus et al. 2018 proved a PAC bound showing that their matrix completion method can recover the subspace of $Z$ w.h.p. (Theorem 2) and hence recover the ATE (Theorem 3). Hence, our model is identifiable if we choose a linear (non-neural parametrization); unfortunately, such linear models do not handle unstructured variables well in practice.
>
> For the general, non-linear setting, finite-sample PAC analysis remains an open question to the best of our knowledge. Wang and Blei’s deconfounder method provides a non-parametric identification result, but they operate under the strong and unverifiable assumption that every (multi-cause) confounder is *pinpointed* by the observed data (Assumption 2). Our method can be framed in this way, in which we assume that our generative model can reconstruct the latent $Z$ using the multi-modal observations.
>
> **Citations**:
>
> Pearl, J. Causality. Cambridge University Press, 2009
>
> Kuroki, Manabu, and Judea Pearl. "Measurement bias and effect restoration in causal inference." Biometrika 101.2 (2014): 423-437.
>
> Kallus, Nathan, Xiaojie Mao, and Madeleine Udell. "Causal inference with noisy and missing covariates via matrix factorization." Advances in neural information processing systems 31 (2018).
>
> Wang, Yixin, and David M. Blei. "Towards clarifying the theory of the deconfounder." arXiv preprint arXiv:2003.04948 (2020).

---

> > ### Author Response · Authors · 2022-08-03
> > **Answering Reviewer Questions**
> >
> > **Answering Remaining Reviewer Questions**
> >
> > Note that we already compare against VAE-based approaches, e.g., in the missing data experiment (Section 5.4) and on GWAS (Section 5.3). The single-modality DSEs are effectively comparable to existing VAE-based methods (e.g., CEVAE), and our multi-modal approach performs better.
> >
> > Examples of proxy variables admissible in our models include:
> > - Unstructured proxy variables: e.g., wearable sensor time series data as a proxy for a patient’s health
> > - Structured proxy variables: e.g., BMI as a proxy of the subject’s health
> > - In the aforementioned extension, we also admit variables V that correspond to observed confounders: e.g., a patient’s smoking status.
> >
> > **Answering specific minor issues/concerns**:
> >
> > *Confounder is not defined in this paper*: We define Z as confounder variable in the background section.
> >
> > *Are "unstructured proxy variables", "unstructured multi-modal proxy variables" and "unstructured data" the same in this work? It is better to keep consistent*.
> >
> > The unstructured data available in modern datasets can be used as proxies to extract hidden confounder during causal effect estimation. This unstructured data can be in the form of multiple modalities (e.g images, tabular data, genomic sequence data, etc). We will explain this better in the final version of the paper.
> >
> > *Line 143 to 147, It would be better to provide a more detailed explanation. The current evidence and conclusion given are somewhat sloppy*.
> >
> > Here, we are using product-of-experts formulation to infer the posterior. We will add a detailed derivation of the factorization in the appendix so that this is clearer. In lines 143 to 147, we point out that we can compute the posterior in closed form when the terms on the right-hand side in Equation 6 are Gaussians. In this special case, it is possible to compute the product of these distributions as another Gaussian with mean and standard deviation as specified in these lines. We will make this clearer and add more explanation on this to the appendix. Please also refer to the derivations provided in **Appendix H**.
> >
> > *The conclusion in Section 5.1 is not clear from the current descriptions.*
> > This section demonstrates a toy example where the causal model (based on our deep structural equations) produces better ATE estimates on a test dataset. We show that substituting a binary input variable with unstructured image modality does not degrade the ATE estimates. Thank you for pointing out that some parts were not clear here - we will address this in the final version of the paper.
> >
> >
> > We thank the reviewer for pointing out the following typos. We will correct these.
> > *multiple multi-model, it is better to remove `multiple'*.
> > *After Line 198, "1", "0" -> ``1", ``0"*

---

> > > ### Author Response · Authors · 2022-08-09
> > > **Additional Experiments**
> > >
> > > Below, we are adding additional experiments and simulations to support our claims. We report __additional experiments__ on two settings: modified causal graph structures and a head-to-head comparison of DMSE vs. CEVAE.
> > >
> > > **Comparing DMSE with CEVAE**
> > >
> > > In our previous response, we identified key differences between DMSE and CEVAE: (1) a modified causal graph architecture that supports multiple unstructured and possibly missing proxies; (2) improved inference algorithms. We now show that these differences lead to differences in performance in practice by performing a simulated experiment. In all settings, we **outperform the popular CEVAE model**.
> > >
> > > First, we generate synthetic data using a process similar to the synthetic experiment already present in our paper (see Appendix G.5 for details). There is only one modality in this experiment; the key difference between the two models in the inference procedure. Table 1 shows that even in this setting, DMSE recovers the ATE more accurately than CEVAE.
> > >
> > > **Table 1: Comparison of DMSE with CEVAE on one modality**
> > > | Model | ATE error (train+val) | ATE error (test) |
> > > | --- | --- | --- |
> > > | CEVAE | 6.37E-02  (1.62E-02) | 6.41E-02 (1.78E-02) |
> > > | DMSE | 3.28E-02 (3.96E-03) | 3.33E-02 (4.49E-03) |
> > >
> > > Next, we compare CEVAE vs. DMSE when the data has **many unstructured modalities** in Table 2. We generate synthetic data from K modalities. The details on this are in **(Appendix G.5) (Dataset E)**. The CEVAE model treats them as one concatenated vector; DMSE models them as separate vectors. As expected, DMSE handles large numbers of modalities better than CEVAE.
> > >
> > > **Table 2: Comparison of DMSE with CEVAE under increasing number of input proxies**
> > > | Number of input modalities | CEVAE | | DMSE | | % improvement made by DMSE w.r.t CEVAE | |
> > > | --- | --- | --- | --- | --- | --- | --- |
> > > | | ATE error (train+val) | ATE error (test) | ATE error (train+val) | ATE error (test) | ATE error (train+val) | ATE error (test) |
> > > | 5 | 0.0533 (0.0165) | 0.0663 (0.0244) | 0.0421 (0.0045) | 0.0472 (0.0166) | 21.0% | 28.8% |
> > > | 10 | 0.0381 (0.0122) | 0.0425 (0.0148) | 0.0296 (0.0040) | 0.0334 (0.0052) | 22.5% | 21.5% |
> > > | 15 | 0.0465 (0.0062) | 0.0545 (0.0112) | 0.0350 (0.0066) | 0.0408 (0.0011) | 24.7% | 25.1% |
> > > | 20 | 0.0764 (0.0178) | 0.0738 (0.0164) | 0.0407 (0.0087) | 0.0383 (0.0054) | 46.6% | 48.1% |

---

### Meta-Review · Area_Chair_h84S · 2022-08-29

**Recommendation:** Accept
**Confidence:** Less certain

**Metareview:**

Reviewers agreed the paper presents a novel method addressing an important problem, building on and expanding prior work in the field. Specifically, it strongly relates to the CEVAE model, adding to it the ability to deal with multiple proxies each with a different structure, as well as introducing a new inference approach. Extensive experimental evaluation (some following the reviews) shows overall strong results. There were concerns with the somewhat limited level of novelty, and lack of theoretical foundations.

NB: The definition of ITE in the paper should in fact be CATE, the Conditional Average Treatment Effect, as it is conditioned on a variable X=x, and not a singe unit's effect.

**Award:**

No

---

### Decision · Program_Chairs · 2022-09-14

Accept